# To Compress or Not? Pushing the Frontier of Lossless GenAI Model Weights Compression with Exponent Concentration

**Zeyu Yang[1], Tianyi Zhang[1], Jianwen Xie[2], Chuan Li[2], Zhaozhuo Xu[3],**
**Anshumali Shrivastava[1]**
[1]Rice University, [2]Lambda, Inc., [3]Stevens Institute of Technology
`{zy45, tz21, as143}@rice.edu`, `{jianwen.xie, c}@lambdal.com`,
`zxu79@stevens.edu`

## Abstract

The scaling of Generative AI (GenAI) models into the hundreds of billions of parameters makes low-precision computation indispensable for efficient deployment. We argue that the fundamental solution lies in developing low-precision *floating-point* formats, which inherently provide numerical stability, memory savings, and hardware efficiency without dequantization overhead. In this paper, we present a theoretical and empirical study of an *exponent concentration* phenomenon in GenAI weights: exponents consistently exhibit low entropy across architectures and modalities. We show that this arises naturally from $\alpha$-stable distributions induced by stochastic gradient descent, and we prove tight bounds on the entropy of exponents. Our analysis establishes a theoretical compression limit near FP4.67, which motivates the design of a practical FP8 format. Building on these insights, we propose **Exponent-Concentrated FP8 (ECF8)**, a lossless compression framework with entropy-aware encoding and GPU-optimized decoding. Experiments on LLMs and DiTs up to 671B parameters demonstrate up to 26.9% memory savings and 177.1% throughput acceleration, with perfectly lossless computations, i.e., no deviation in model outputs. Our results establish exponent concentration as a statistical law of trained models and open a principled path for lossless low-precision floating-point design in the FP8 era. Code is available at `https://github.com/zeyuyang8/ecf8`.

## 1 Introduction

The rapid growth of generative AI (GenAI) models, from large language models (LLMs) (Dubey et al., 2024; Team et al., 2023; Hurst et al., 2024; Liu et al., 2024a; Yang et al., 2025) to diffusion transformers (DiTs) (Wan et al., 2025; Batifol et al., 2025; Wu et al., 2025), has led to parameter counts scaling into the hundreds of billions. Such parameter-intensive GenAI models make low-precision computation indispensable. General matrix multiplication (GeMM) (Dongarra et al., 1990; Goto & Geijn, 2008; Springer & Bientinesi, 2018) with reduced precision is the most straightforward way to achieve memory savings and possible acceleration, as it eliminates redundancy in weight representation while directly reducing hardware workloads.

Existing work has primarily focused on integer-based quantization (Zhang et al., 2024; Zhang & Shrivastava, 2024; Yao et al., 2022; Dettmers et al., 2022; Frantar et al., 2022; Xiao et al., 2023; Lin et al., 2024a; Liu et al., 2024b). While effective in reducing model size, these approaches suffer from two fundamental drawbacks: (a) they are *lossy*, often introducing accuracy or generative quality degradation (Zhang et al., 2025); and (b) they incur efficiency penalties in large-batch inference due to the required dequantization procedure and mixed-precision execution (Lin et al., 2024b; Jin et al., 2024). Since integer tensors must be converted back into floating-point values before computing, throughput suffers, limiting their utility in high-performance deployment.

Motivated by the limitations of lossy quantization, DFloat11 (Zhang et al., 2025) revealed that the exponents of BF16 weights in LLMs have far lower entropy than the bitwidth allocated to store them,

creating headroom for lossless compression via entropy coding. Heilper & Singer (2025) reported similar findings about low-entropy exponents in neural networks. Yet these findings puzzled the community: Is there a fundamental principle that explains its success? Can the idea extend beyond BF16 to other formats? What is the fundamental lower bound on exponent entropy, and how does it inform the design of future numerical formats for neural networks? And most critically, can memory compression be transformed into actual *inference acceleration* by getting fast decoding kernels? All of these unanswered questions can be answered positively by our paper.

In this paper, we present a theoretical and empirical analysis revealing an *exponent concentration* phenomenon in GenAI weights: exponents exhibit low entropy (Shannon, 1948) and cluster within narrow ranges across architectures and modalities. We trace this to the heavy-tailed dynamics of stochastic gradient descent (Amari, 1993), which lead neural network weights to follow $\alpha$-stable distributions (Nikias & Shao, 1995). This provides a rigorous explanation of why exponents concentrate and allows us to bound their entropy. Our analysis proves that the ultimate compression limit corresponds to a floating-point format of roughly FP4.67. While such a fractional format is impractical in modern GPU hardware, it motivates our design of a powerful, lossless FP8 variant.

Building on these insights, we introduce **Exponent-Concentrated FP8 (ECF8)**, a novel lossless compression framework that encodes exponents with entropy-aware coding and efficient GPU decoding. We show that *ECF8 can transform memory compression into inference acceleration*, delivering up to 26.9% memory savings and up to 177.1% throughput acceleration across state-of-the-art GenAI models, without any observed deviation in generation. ECF8 establishes a principled path toward the theoretical limit of lossless weight compression in neural networks. To the best of our knowledge, ECF8 is the first lossless compression method that delivers end-to-end inference acceleration and is validated across GenAI models up to 671B parameters.

In summary, our contributions are:

- We provide a theoretical analysis of exponent concentration in GenAI weights, proving that $\alpha$-stable distributions lead to bounded low-entropy exponents with a compression limit near FP4.67.

- We empirically validate exponent concentration across large LLMs and DiTs, showing that layer-wise entropy consistently lies around 2–3 bits.

- We design and implement **ECF8**, a lossless FP8 compression framework with encoding and GPU-optimized decoding, leading to inference acceleration.

- We demonstrate practical benefits on models up to 671B parameters, achieving significant memory reduction and throughput gains while preserving bit-exact fidelity.

## 2 EXPONENT CONCENTRATION LEADS TO LOSSLESS WEIGHT COMPRESSION IN GENERATIVE AI

**Preliminary.** Floating-point numbers in IEEE-754 format consist of three components: a sign bit $s$, an exponent $E$, and a mantissa $M$. In the low-precision computing paradigm of deep learning, the exponents determine the dynamic range of representable values. If the exponent values are highly concentrated rather than spread uniformly, their entropy is low, which directly implies that fewer bits are needed for lossless representation. This motivates analyzing the statistical laws that govern exponent distributions in trained model weights.

### 2.1 OBSERVATION: LOW-ENTROPY EXPONENTS IN GENERATIVE AI MODEL WEIGHTS

We empirically examine weight matrices from parameter-intensive GenAI models, including transformers used for language and vision tasks. Across diverse architectures (LLMs, diffusion models, and multimodal transformers), we consistently observe that the exponents of weight values occupy a very narrow range and have much lower entropy than the allocated bitwidth would suggest. As shown in Figure 1, histograms of exponent values cluster tightly around a few modes, and the measured Shannon entropy is typically close to 2-3 bits, in contrast to the 8 or more bits allocated in standard floating-point formats. This persistent low-entropy phenomenon strongly suggests an underlying distributional principle.

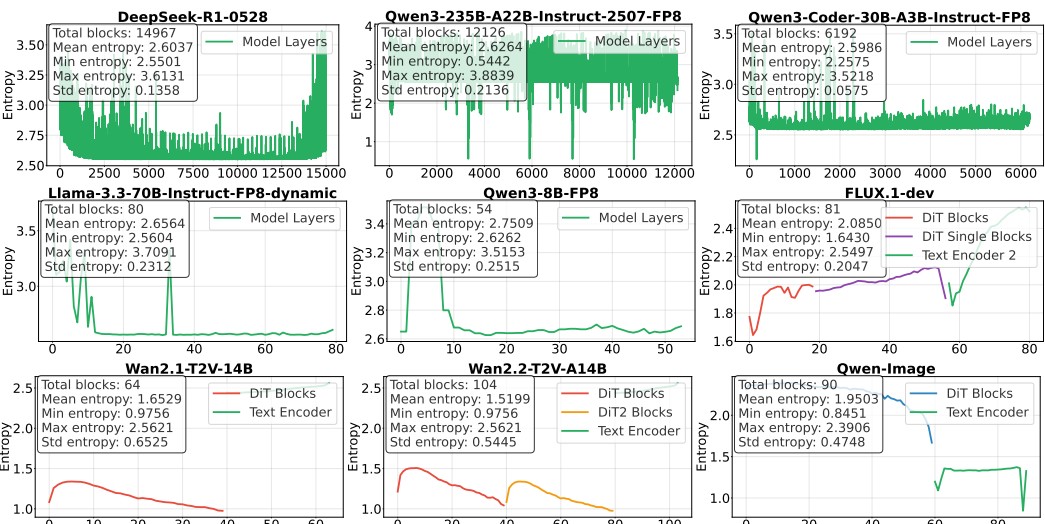

Figure 1: Entropy analysis across transformer blocks for different model architectures. The x-axis represents the block index within each model, and the y-axis shows the entropy values. Different colors indicate different block types within each architecture.

## 2.2 ANALYSIS: EXPONENT CONCENTRATION OF $\alpha$-STABLE VARIABLES

**Set-up and notations.** Let $X \in \mathbb{R}$ be a continuous random variable representing a neural network weight. In the IEEE-754 floating-point representation, a nonzero real number $x$ is encoded as:

$$x = (-1)^s \cdot 2^E \cdot M, \quad \text{with } E \in \mathbb{Z}, M \in [1, 2),$$

where $s$ is the sign bit, $E$ is the exponent, and $M$ is the mantissa. We define:

$$E = \lfloor \log_2 |X| \rfloor,$$

as the *floating-point exponent* of $X$. Our aim is to analyze the entropy and compression limit of $E$ when $X$ follows a symmetric $\alpha$-stable distribution:

$$X \sim S_\alpha(\beta = 0, \gamma, \delta), \quad \text{with } \alpha \in (0, 2].$$

### 2.2.1 WHY NEURAL NETWORK WEIGHTS FOLLOW $\alpha$-STABLE DISTRIBUTIONS

Trained neural network weights are formed by accumulating stochastic updates over time. Under stochastic gradient descent (SGD), the update rule $\theta_{t+1} = \theta_t - \eta \cdot \nabla \mathcal{L}(\theta_t; \xi_t)$ introduces heavy-tailed noise from random mini-batch sampling. Empirical work shows that the gradient noise often exhibits power-law tails $\mathbb{P}(|\Delta_t| > x) \sim x^{-\alpha}$ with $\alpha < 2$. By the Generalized Central Limit Theorem, sums of such heavy-tailed variables converge to $\alpha$-stable distributions. Thus, neural network weights after many updates approximately follow symmetric $\alpha$-stable laws, providing the theoretical foundation for our analysis.

### 2.2.2 EXPONENT ENTROPY CONCENTRATION

**Theorem 2.1** (Exponent Entropy Concentration). *Let $E = \lfloor \log_2 |X| \rfloor$ where $X \sim S_\alpha(\beta = 0, \gamma, \delta)$. Then $E$ follows a discrete two-sided geometric distribution with parameter $q = 2^{-\alpha}$:*

$$\mathbb{P}(E = k) = \frac{1 - q}{1 + q} \cdot q^{|k|}, \quad k \in \mathbb{Z}.$$

*The Shannon entropy of $E$ is bounded by:*

$$\frac{\alpha}{1 + 2^{-\alpha}} \le H(E) \le \frac{\alpha}{1 - 2^{-\alpha}}.$$

*In particular, $H(E)$ is finite for all $\alpha > 0$.*

*Proof.* For large $|x|$, the tails of an $\alpha$-stable distribution behave like $\mathbb{P}(|X| > x) \sim C_\alpha x^{-\alpha}$. Thus, the probability of exponent $E = k$ is:

$$\mathbb{P}(E = k) \approx \mathbb{P}(2^k \leq |X| < 2^{k+1}) = C_\alpha 2^{-k\alpha}(1 - 2^{-\alpha}),$$

which corresponds to a two-sided geometric distribution. The entropy of this distribution is:

$$H(E) = -\sum_{k \in \mathbb{Z}} \mathbb{P}(E = k) \log_2 \mathbb{P}(E = k) = h_2\left(\frac{1-q}{1+q}\right) + \frac{2q}{1+q} \cdot \frac{|\log_2 q|}{1-q},$$

where $h_2(p) = -p \log_2 p - (1-p) \log_2(1-p) \leq 1$ is the binary entropy. Plugging $q = 2^{-\alpha}$ and bounding terms gives the inequality. $\square$

**Interpretation.** This theorem shows that the floating-point exponents of $\alpha$-stable weights do not spread evenly across the integer line but instead concentrate around zero, decaying geometrically with rate $2^{-\alpha}$. The entropy bound demonstrates that this concentration is strong enough to guarantee finite entropy regardless of $\alpha$, and tighter concentration (smaller $\alpha$) leads to smaller entropy. Practically, this means that exponents carry very limited uncertainty, which enables efficient compression.

## 2.3 IMPACT: EXPONENT CONCENTRATION GUIDES LOSSLESS COMPRESSION

**Corollary 2.2** (Compression Limit). *The minimal expected number of bits required to losslessly encode the exponent $E$ is:*

$$L_{\min} = H(E).$$

*Therefore, exponent values of neural network weights drawn from an $\alpha$-stable distribution can be encoded in:*

$$O\left(\frac{\alpha}{1 - 2^{-\alpha}}\right) \text{ bits on average.}$$

**Numerical instance** ($\alpha = 2$). When $\alpha = 2$ (the Gaussian-like case), we have $2^{-\alpha} = 1/4$. The entropy bounds give:

$$1.6 \leq H(E) \leq 2.67.$$

Thus, the exponent itself has a compression limit of about 2.67 bits in the extreme case. However, a floating-point representation also requires one sign bit and several bits for the mantissa to preserve numerical precision. Even with a minimal mantissa allocation (e.g., $\sim 1$ bit) plus the sign, the absolute floor is around:

$$2.67 + 1 + 1 \approx 4.67 \text{ bits.}$$

In practice, it is infeasible to implement a "FP4.67" or even FP5 format efficiently due to alignment and hardware constraints. Therefore, our proposed FP8 format (ECF8) represents a practical engineering choice: it is close to the entropy-driven theoretical limit, while retaining sufficient mantissa precision and hardware compatibility for efficient inference.

# 3 ECF8: LOSSLESS LLM WEIGHT COMPRESSION WITH EXPONENT CONCENTRATION

We present ECF8, a lossless compression algorithm that exploits the statistical properties of FP8 weights in pre-trained generative AI models. Our method consists of three core components: an encoding scheme based on Huffman coding, a parallel GPU decoding kernel for variable-length decoding, and a dynamic tensor management system that reduces memory footprint during inference.

## 3.1 ENCODING

Our CUDA-based Huffman encoding pipeline transforms neural network weights into a compressed format optimized for parallel GPU decoding through three sequential stages. First, we generate optimal Huffman codes by analyzing weight exponent frequencies and constructing the corresponding binary tree. Second, we build hierarchical lookup tables that enable efficient variable-length code decoding using 8-bit subtables aligned with GPU memory architecture. Third, we encode weight exponents into a compressed bitstream while computing synchronization metadata that enables autonomous parallel decoding across GPU thread blocks.

**Huffman code generation.** Neural network weights in FP8 format allocate 4 bits for exponents, yet empirical analysis reveals that exponent entropy is substantially lower due to non-uniform frequency distributions. We exploit this entropy gap through Huffman coding, which assigns variable-length prefix-free codes with shorter sequences for frequent exponents. Our encoding procedure extracts exponents from model weights, computes their empirical frequency distribution $p(x)$ for exponent values $x \in \{0, 1, \ldots, 15\}$, and constructs the optimal Huffman tree minimizing expected code length $\mathbb{E}[\ell] = \sum_x p(x)\ell(x)$. To ensure GPU compatibility, we constrain maximum code length to 16 bits, requiring frequency adjustment for rare symbols while preserving near-optimality. This constraint is rarely violated in transformer layers empirically.

**Hierarchical lookup table construction.** We construct a multi-level lookup system that processes variable-length Huffman codes through sequential 8-bit operations. The system comprises two components: a *cascaded decode table* mapping codes to symbols, and a *length table* storing code lengths indexed by symbol value.

The cascaded structure organizes codes by their byte-aligned prefixes. Let $\mathcal{P} = \{p_1, p_2, \ldots, p_k\}$ denote the set of all byte-aligned prefixes extracted from Huffman codes, ordered by length. For each prefix $p_i$ of length $8j$ bits, we construct a lookup subtable $LUT_i$ with 256 entries. Each entry $LUT_i[b]$ for byte value $b \in \{0, \ldots, 255\}$ contains either

- a decoded exponent $x \in \{0, \ldots, 15\}$ if the code $p_i \| b$, where $\|$ denotes concatenation, corresponds to a complete Huffman code,
- or a pointer value $256 - \text{index}(p_{i'})$ directing lookup to subtable $LUT_{i'}$ for longer prefix $p_{i'} = p_i \| b$.

This design bounds memory usage at $O(|\mathcal{P}| \cdot 256)$ entries while maintaining $O(\lceil \ell_{\max}/8 \rceil)$ lookup time for codes of length $\ell_{\max}$. The length table $L[x]$ stores the bit length of each symbol $x$, enabling proper bitstream advancement during decoding. Figure 2 illustrates the construction process of the *cascaded decode table* using a simplified example using the characters "a", "b", "c", "d", and "e" as symbols rather than exponents $x \in \{0, \ldots, 15\}$.

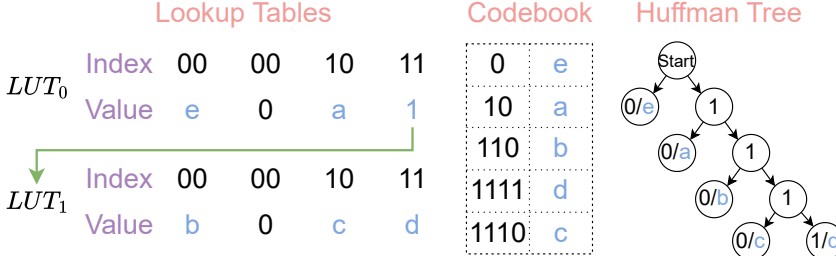

Figure 2: A simplified illustration of lookup table construction. A Huffman tree is built from the string "aaabbcddeeee". Lookup tables are configured to be 2-bit, so each subtable has 4 entries. Codes for symbols "e" and "a" are at most 2 bits long and appear directly in the first table. In contrast, codes for "b", "c", and "d" exceed 2 bits and begin with "11", so entry "11" in the first table points to a secondary table. The pointer value is 1, indicating subtable 1. A second lookup in this subtable resolves the final symbol.

**Encoding and synchronization metadata generation.** We encode exponents into a compressed bitstream while generating coordination metadata for parallel GPU decoding. Given fixed parameters $B$ (bytes per thread) and $T$ (threads per block), we sequentially process symbols $x_1, x_2, \ldots, x_n$, concatenating their Huffman codes into a continuous bitstream and extracting complete bytes for storage. Since variable-length codes may span thread boundaries, we compute *gap values* to ensure proper synchronization. For thread $t$ processing bytes $[tB, tB + 2)$, the gap $g_t$ represents the bit offset from the previous thread:

$$g_t = \left( \sum_{i=0}^{t-1} \sum_{x_j \in \text{symbols}(i)} \ell(x_j) \right) \bmod 8B$$

where $\text{symbols}(i)$ denotes the set of symbols processed by thread $i$. Additionally, we compute *output positions* $o_b$ for each thread block $b$, tracking the cumulative count of symbols processed

by preceding blocks. This metadata enables autonomous thread block operation during parallel decompression without requiring inter-block synchronization.

## 3.2 DECODING

The decoding process reconstructs original exponent values from the Huffman-encoded bitstream using a highly optimized CUDA kernel that exploits GPU parallelism through hierarchical memory management and coordinated thread execution. The decoding algorithm operates in five distinct phases: **i. memory initialization**, where each thread allocates a register buffer of fixed size (bytes per thread plus additional lookahead bytes) and the thread block establishes shared memory for coordination and output staging; **ii. data loading**, where threads load their assigned segments of the encoded bitstream from global memory into thread-local registers; **iii. parallel counting**, where each thread performs initial decoding based on gap values to determine the number of decodable symbols, followed by a parallel reduction algorithm across the thread block to compute cumulative symbol counts and establish output positions for each thread (ensuring threads write to non-overlapping memory regions); **iv. coordinated decoding**, where threads decode their full symbol sequences and write results to shared memory using the previously computed output positions; and **v. global memory write-back**, where decoded symbols are transferred from shared memory to global memory via coalesced writes. This approach minimizes global memory access overhead while maximizing parallelism through careful coordination of thread-local computation and block-level synchronization primitives. A detailed description of the decoding algorithm is provided in Algorithm 1 in Appendix M.

## 3.3 TENSOR MANAGEMENT

Our tensor management system enables on-demand weight decompression during model inference through strategic memory allocation and layer-wise processing. We implement a just-in-time decompression mechanism using PyTorch forward hooks that intercept layer execution and perform weight reconstruction immediately before computation. The system maintains a single pre-allocated GPU memory buffer of size equal to the largest layer's weight tensor, eliminating dynamic memory allocation overhead during inference. For each layer $\ell_i$ in the sequential model architecture, the forward hook invokes our CUDA decompression kernel to reconstruct weights $W_i$ from their compressed representation and writes the result to the shared memory buffer. Upon completion of layer $\ell_i$'s forward pass, the buffer becomes available for layer $\ell_{i+1}$, enabling memory-efficient inference with constant GPU memory overhead regardless of model depth. Note that ECF8 is designed to optimize inference performance. Appendix G shows how ECF8 can be integrated with popular inference frameworks such as vLLM. Appendix H explains why ECF8 is compatible with LoRA (Hu et al., 2022).

## 4 EXPERIMENTS

In this section, we test nine models spanning autoregressive language models, diffusion transformers, and mixture-of-experts variants from 8B to 671B parameters. We evaluate ECF8 across two critical dimensions: compression effectiveness and end-to-end inference performance. Specifically, we aim to address the following two fundamental research questions related to the deployment of production-scale GenAI models:

- **RQ1:** What memory reduction can production-scale transformers achieve while maintaining bit-exact weight reconstruction?

- **RQ2:** How does weight compression affect memory footprint and inference latency and can we achieve faster inference under the same memory budget?

## 4.1 LOSSLESS MEMORY SAVING FOR FP8 WEIGHTS

This section addresses **RQ1** regarding the rate of memory reduction for production-scale transformers. ECF8 achieves memory reductions from 9.8% to 26.9% across all evaluated models. LLMs demonstrate consistent reductions between 9.8% and 14.8%, while DiTs achieve higher compres-

sion ratios, with the Wan2.2-T2V-A14B model reaching 26.9% reduction. Figure 3 shows that the compression is lossless.

Table 1: Memory savings and end-to-end throughput improvements under fixed memory constraints. For throughput evaluation, DeepSeek-R1-0528 is tested on 8×H200 systems, Qwen3-235B-A22B-Instruct-2507-FP8 on 4×H200 systems, and the remaining models on single GH200 (96GB) systems. The "Supported Machine" column indicates the minimum hardware configuration capable of running the ECF8-compressed model. Memory savings and throughput improvements under fixed memory constraints demonstrate ECF8's practical deployment benefits. Appendix C provides the entropy analysis for the evaluated models, while Appendix I summarizes the ECF8 encoding time.

| Model | Memory Change (GB) | Memory ↓ (%) | Supported Machine | Throughput ↑ (%) |
|---|---|---|---|---|
| DeepSeek-R1-0528 | 623.19 → 530.26 | 14.8 | 8×H100 (80 GB) | 150.3 |
| Qwen3-235B-A22B-Instruct-2507-FP8 | 217.77 → 185.98 | 14.4 | 4×H100 (80 GB) | 35.9 |
| Llama-3.3-70B-Instruct-FP8-dynamic | 63.76 → 54.69 | 13.4 | 1×H100 (80 GB) | 11.3 |
| Qwen3-Coder-30B-A3B-Instruct-FP8 | 27.85 → 23.69 | 14.3 | 1×RTX5090 (32 GB) | 23.7 |
| Qwen3-8B-FP8 | 6.47 → 5.61 | 9.8 | 1×RTX4070 (12 GB) | 12.6 |
| FLUX.1-dev | 10.52 → 8.29 | 14.1 | 1×RTX4070 (12 GB) | 177.1 |
| Wan2.1-T2V-14B | 17.40 → 12.65 | 25.4 | 1×RTX4080 (16 GB) | 55.1 |
| Wan2.2-T2V-A14B | 30.49 → 21.85 | 26.9 | 1×RTX4090 (24 GB) | 108.3 |
| Qwen-Image | 26.20 → 20.56 | 21.0 | 1×RTX4090 (24 GB) | 126.6 |

ECF8 delivers memory reductions from 9.8% to 26.9% across all evaluated models, with effectiveness correlating strongly with architecture type. Language models achieve moderate but consistent reductions between 9.8% and 14.8%, while diffusion transformers show substantially higher compression potential, with the Wan2.2-T2V-A14B model reaching 26.9% reduction.

This pattern validates our theoretical foundation: FP8 exponent distributions in trained networks contain exploitable redundancy through lossless compression. Compression effectiveness remains remarkably stable across model scales, from 8B parameter Qwen3-8B-FP8 to 671B parameter DeepSeek-R1-0528, indicating that ECF8's performance depends on fundamental weight distribution properties rather than model size.

The practical impact extends far beyond storage efficiency. Memory reductions enable deployment on lower-capacity hardware: the 14.8% reduction for DeepSeek-R1-0528 allows 8×H100 deployment instead of requiring 8×H200 or 2-node 8×H100 systems, while the 25.4% reduction for Wan2.1-T2V-14B fits within single RTX4080 constraints where uncompressed models exceed memory limits.

Under fixed memory constraints, ECF8 enables throughput improvements ranging from 11.3% to 177.1% by supporting larger batch sizes within the same memory budget. These gains compound ECF8's benefits beyond simple storage efficiency to deliver measurable inference performance improvements. Details of inference speed evaluation are presented in Section 4.2.

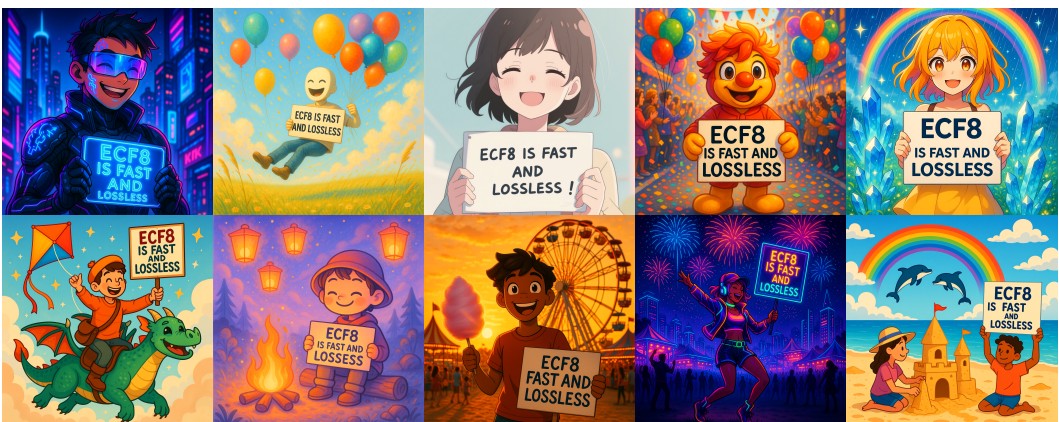

Figure 3: Images generated by ECF8-compressed Qwen-Image model, demonstrating pixel-perfect reconstruction quality compared to the original FP8 model. The numerical difference between images generated by ECF8 model and FP8 model using the same random seed and inference parameters is perfectly zero.

Figure 3 provides direct evidence of ECF8's lossless compression property through visual quality assessment. The ECF8-compressed Qwen-Image model generated these images using identical random seeds and inference parameters as the uncompressed FP8 baseline. Each output is pixel-by-pixel identical to the original model's results, confirming zero quality degradation. Appendix D also demonstrates the lossless nature of ECF8 through numerical evaluations of compressed LLMs on standard downstream language tasks.

This lossless property proves critical for production deployment, where model behavior must remain completely unchanged while achieving significant memory savings. The diverse generated content demonstrates that ECF8 preserves full generative capabilities across different image domains and styles without introducing any artifacts or quality loss.

## 4.2 IMPROVEMENT ON INFERENCE SPEED UNDER FIXED MEMORY FOOTPRINT

This section addresses **RQ2** regarding the inference acceleration under a fixed memory footprint. We evaluate inference speed improvements by measuring maximum achievable throughput within fixed memory budgets, simulating real-world scenarios where hardware capacity limits batch sizes. In conclusion, the memory savings achieved by ECF8 translate directly into substantial inference performance improvements under realistic deployment constraints. The rationale behind this observation is that smaller memory footprints allow for larger batch sizes, which in turn enable higher throughput, as well as shorter per-request latency. For instance, under a 640 GB memory limit, ECF8 DeepSeek-R1-0528 sustains a batch size of 16, delivering 150.3% higher throughput and 60.1% lower per-request latency than FP8, which can only accommodate a batch size of 2. In addition, Appendix E provides a detailed analysis of the kernel-level performance of ECF8 decompression and Appendix F compares the throughput of FP8 and ECF8 LLMs at the same batch size.

Table 2: Comparison of FP8 and ECF8 across LLMs of varying scales. Under fixed memory constraints, we report per-request latency (s), throughput (tokens/s), and maximum batch size (each batch generates 1024 tokens). DeepSeek-R1-0528 and Qwen3-235B-A22B-Instruct-2507 are evaluated on H200 GPUs (141 GB), while other models are evaluated on a single GH200 GPU (96 GB).

| Model / Constraint | | Max Batch Size | | Per Request Latency (s) | | | Throughput (tokens/s) | | |
|---|---|---|---|---|---|---|---|---|---|
| Model | Constraint | FP8 | ECF8 | FP8 | ECF8 | ↓ (%) | FP8 | ECF8 | ↑ (%) |
| DeepSeek-R1-0528 | 640 GB | 2 | 16 | 660.65 | 263.95 | 60.1 | 1.55 | 3.88 | 150.3 |
| Qwen3-235B-A22B-Instruct-2507-FP8 | 240 GB | 32 | 64 | 107.56 | 79.14 | 26.4 | 9.52 | 12.94 | 35.9 |
| Llama-3.3-70B-Instruct-FP8-dynamic | 80 GB | 32 | 48 | 24.80 | 22.28 | 10.2 | 41.28 | 45.96 | 11.3 |
| Qwen3-Coder-30B-A3B-Instruct-FP8 | 32 GB | 16 | 32 | 107.33 | 86.70 | 19.2 | 9.54 | 11.80 | 23.7 |
| Qwen3-8B-FP8 | 12 GB | 16 | 24 | 4.90 | 4.35 | 11.2 | 208.80 | 235.22 | 12.6 |

**Language model inference acceleration.** Table 2 results demonstrate consistent throughput improvements across language models ranging from 11.3% to 150.3%. The DeepSeek-R1-0528 model shows the most dramatic improvement, achieving 150.3% higher throughput (3.88 vs 1.55 tokens/s) by enabling $8\times$ larger batch sizes (16 vs 2) within the 640 GB memory constraint. This improvement stems from ECF8's ability to reduce the model's memory footprint from 623GB to 530GB, creating sufficient headroom for larger batches.

Smaller models exhibit more modest but still significant gains. The Qwen3-8B-FP8 model achieves 12.6% throughput improvement by increasing batch size from 16 to 24 within a 12GB constraint. Even with constrained memory budgets typical of consumer hardware, ECF8 consistently enables meaningful performance improvements through more efficient memory utilization.

**Diffusion model inference acceleration.** Diffusion models are primarily compute-bound rather than memory-bound, with latency dominated by extensive denoising computations across multiple timesteps. VRAM management, which dynamically offloads and reloads model components between GPU and CPU memory, represents a common deployment strategy for diffusion models to handle memory constraints during inference. As shown in Table 3, ECF8 consistently outperforms FP8 baselines across four representative diffusion transformer architectures, achieving memory reductions from 7.9% to 17.8% and latency improvements from 3.3% to 55.9% under controlled experimental conditions using the DiffSynth library. FLUX.1-dev exhibits the most substantial gains with 45.9% end-to-end latency reduction (13.15s vs 24.29s) and 12.1% memory savings, while Qwen-

Table 3: Comparison of FP8 and ECF8 across popular DiTs. Reported metrics include end-to-end (E2E) latency (s), step latency (ms), and GPU peak memory (MB). All experiments are conducted with the DiffSynth library on a single GH200 GPU (96 GB) using an identical batch size, random seed, and prompt. DiffSynth enables VRAM management by default, dynamically offloading and reloading model components between GPU and CPU to reduce peak memory usage.

| Model | DType | E2E Latency (s) | Step Latency (ms) | Memory (MB) | Memory ↓ (%) | Latency ↓ (%) |
|---|---|---|---|---|---|---|
| FLUX.1-dev | ECF8 | $13.15 \pm 0.08$ | $438.4 \pm 2.8$ | 14274 | 12.1 | 45.9 |
| | FP8 | $24.29 \pm 0.10$ | $809.5 \pm 3.4$ | 16243 | 0.0 | 0.0 |
| Wan2.1-T2V-14B | ECF8 | $460.67 \pm 0.92$ | $9213.4 \pm 18.5$ | 18036 | 7.6 | 3.3 |
| | FP8 | $476.21 \pm 3.34$ | $9524.3 \pm 66.8$ | 19529 | 0.0 | 0.0 |
| Wan2.2-T2V-A14B | ECF8 | $461.41 \pm 1.06$ | $9228.2 \pm 21.3$ | 27560 | 17.8 | 4.0 |
| | FP8 | $480.45 \pm 0.68$ | $9608.9 \pm 13.5$ | 33517 | 0.0 | 0.0 |
| Qwen-Image | ECF8 | $49.05 \pm 0.07$ | $1226.3 \pm 1.7$ | 25766 | 7.9 | 55.9 |
| | FP8 | $111.14 \pm 1.39$ | $2778.4 \pm 34.8$ | 27963 | 0.0 | 0.0 |

Image demonstrates exceptional step-level efficiency with 55.9% per-step latency improvement (1226.3ms vs 2778.4ms). The video generation models Wan2.1-T2V-14B and Wan2.2-T2V-A14B show more modest latency improvements of 3.3-4.0%, though Wan2.2-T2V-A14B achieves 17.8% memory savings. These results validate that ECF8's compact weight representation reduces communication overhead during the frequent weight loading operations characteristic of VRAM-managed diffusion inference, translating storage efficiency into measurable performance gains across diverse model architectures and scales. Beyond single-batch performance gains, the reduced peak memory consumption enables larger batch sizes within fixed memory constraints, translating to remarkable throughput improvements of 55.1% to 177.1% as demonstrated in Table 1.

## 5 RELATED WORK

**Weight quantization of GenAI models.** Quantization (Hubara et al., 2018) has become the dominant compression paradigm, which reduces memory by converting 16-bit weights to lower precision formats. LLM.int8() (Dettmers et al., 2022) achieved practical 8-bit inference through mixed-precision matrix multiplication, and delivered $2\times$ memory reduction by processing most operations in INT8 while it handled outliers in FP16. SmoothQuant (Xiao et al., 2023) enabled W8A8 quantization by migration of activation outliers into weights, while GPTQ (Frantar et al., 2022), AWQ (Lin et al., 2024a), and SpinQuant (Liu et al., 2024b) pushed boundaries to 4-bit precision through second-order optimization, activation-aware selection, and learned rotations. However, quantization is inherently lossy and can cause unpredictable performance degradation, which translates to significant revenue losses at scale. DFloat11 (Zhang et al., 2025) addressed this by exploiting the low entropy of BF16 weights, and proposed lossless compression via entropy-coded dynamic-length floats with efficient GPU decompression, which achieved 30% memory reduction with bit-exact reconstruction. Nevertheless, these methods are specifically designed for BF16 weights. With the emergence of native FP8 models (Micikevicius et al., 2022) as the future trend in GenAI, there remains an unaddressed need for efficient lossless compression techniques tailored to FP8 weights, which motivates our work.

**Weight distribution analysis of neural networks.** A growing body of work indicates that the weight matrices of deep neural networks are not well captured by classical Gaussian assumptions but instead exhibit heavy-tailed behavior that can often be modeled by $\alpha$-stable distributions. On the optimization side, Gurbuzbalaban et al. (2021) showed that stochastic gradient descent gives rise to heavy-tailed dynamics, where the properly rescaled iterates converge in distribution to $\alpha$-stable random variables, with heavier tails (smaller $\alpha$) linked to improved generalization. From the perspective of infinite-width limits, Jung et al. (2021) proved that networks with heavy-tailed initializations converge to $\alpha$-stable processes, while Lee et al. (2023) extended these results to architectures with dependent weights, showing that heavy tails also induce sparsity and compressibility. On the spectral side, Mahoney & Martin (2019) and Martin & Mahoney (2021) analyzed trained weight matrices and observed power-law spectral densities, $\rho(\lambda) \propto \lambda^{-\alpha}$, with most exponents in the range $\alpha \in (2, 4)$ across diverse architectures. Taken together, these findings suggest that heavy-tailed statistics are a robust and recurring feature of well-trained networks. As we show in Section 2,

$\alpha$-stable distributions lead to exponent concentration and, consequently, low entropy, offering opportunities for lossless compression via entropy coding.

# 6 CONCLUSION

We revisited the problem of efficient model deployment for generative AI systems through the lens of low-precision computation. While integer quantization has been the prevailing approach, its lossy nature and reliance on dequantization limit its scalability in high-throughput environments. We demonstrated that neural network weights obey an *exponent concentration* principle, rooted in $\alpha$-stable dynamics of stochastic gradient descent, which ensures that exponents carry bounded low entropy. This theoretical insight yields a fundamental compression limit around FP4.67. Motivated by this, we developed **ECF8**, a practical, lossless FP8 format with entropy-aware encoding and GPU-efficient decoding. Our experiments across LLMs and diffusion transformers show that ECF8 reduces memory consumption by up to 26.9% and improves throughput by as much as 177.1%, all while preserving bit-exact fidelity. Beyond immediate systems benefits, this work highlights exponent concentration as a universal property of trained models and lays the foundation for principled design of next-generation low-precision floating-point formats for GenAI.

## ACKNOWLEDGMENTS

We gratefully acknowledge the support of Lambda, Inc. for providing compute resources for this project. The work of Zhaozhuo Xu was supported by National Science Foundation awards 2451398 and 2450524. The work of Zeyu Yang and Anshumali Shrivastava was supported by Rice Ken Kennedy Institute (K2I) Generative AI Cluster Funding. The work of Anshumali Shrivastava was additionally supported by National Science Foundation award 2336612.

## ETHICS STATEMENT

This work focuses on improving the efficiency of generative AI systems. We do not anticipate any direct negative ethical implications. On the contrary, our approach has the potential to reduce the energy consumption and carbon footprint of AI deployment by lowering GPU requirements. Furthermore, by improving accessibility and reducing computational costs, our work can help democratize the use of generative AI technologies for a broader range of users.

## REPRODUCIBILITY STATEMENT

We have taken several steps to facilitate reproducibility. An open-source GitHub repository containing the full source code and experiment scripts is available at `https://github.com/zeyuyang8/ecf8`. All theoretical results are accompanied by detailed proofs in Section 2. Together, these resources ensure that our results can be independently verified and extended.

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

APPENDIX

## A  THE USE OF LARGE LANGUAGE MODELS (LLMS)

We used LLMs as a general-purpose assistive tool during the preparation of this work. Specifically, LLMs were employed for:

- **Writing assistance:** refining grammar, improving clarity of sentences, and suggesting alternative phrasings without altering the technical content.
- **Formatting:** generating LaTeX code snippets for tables, figures, and algorithm blocks, which were then verified and adapted by the authors.
- **Brainstorming:** offering initial organizational structures for sections and summarizing related work, which the authors subsequently reviewed, fact-checked, and rewrote.

No parts of the research methodology, experiments, data analysis, or conclusions were generated by LLMs. The authors take full responsibility for the final content of the paper. LLMs were not used for creating or fabricating research results. As per ICLR policy, LLMs are not listed as authors.

## B  GENERATIVE AI MODELS USED FOR EVALUATION

We evaluate a diverse set of state-of-the-art generative AI models spanning language, code, image, and video modalities. All models are released in FP8 or have community-supported FP8 versions.

### B.1  LLMS

- DeepSeek-R1-0528[1]: a 671B reasoning-focused mixture-of-experts (MoE) model, one of the first native FP8 LLMs, achieving strong performance on complex reasoning tasks.
- Qwen3-235B-A22B-Instruct-2507-FP8[2]: a 235B MoE FP8 model tuned for instruction following, reasoning, mathematics, coding, and tool use.
- Llama-3.3-70B-Instruct-FP8-dynamic[3]: a 70B dense instruction-tuned model using symmetric FP8 quantization for weights and activations, improving throughput and reducing memory cost.
- Qwen3-Coder-30B-A3B-Instruct-FP8[4]: a 30.5B MoE FP8 model specialized for code generation and agentic tool use, supporting long contexts up to 256K tokens (extendable to 1M with Yarn).
- Qwen3-8B-FP8[5]: an 8B dense FP8 model released by Qwen, serving as a lightweight but competitive baseline.

### B.2  DITS

- FLUX.1-dev[6]: a 16B DiT for text-to-image generation, originally in BF16 with community FP8 implementations.
- Wan2.1-T2V-14B[7]: a 14B DiT for text-to-video generation, BF16 release with FP8 support by community.
- Wan2.2-T2V-A14B[8]: a 30B MoE DiT for text-to-video generation, representing a major architectural upgrade. Released in BF16 with FP8 support by community.
- Qwen-Image[9]: a 20B DiT-based model for image generation and editing, notable for high-fidelity text rendering and precise editing. Released in BF16 with FP8 implementations by community.

---

[1]https://huggingface.co/deepseek-ai/DeepSeek-R1-0528
[2]https://huggingface.co/Qwen/Qwen3-235B-A22B-Instruct-2507-FP8
[3]https://huggingface.co/RedHatAI/Llama-3.3-70B-Instruct-FP8-dynamic
[4]https://huggingface.co/Qwen/Qwen3-Coder-30B-A3B-Instruct-FP8
[5]https://huggingface.co/Qwen/Qwen3-8B-FP8
[6]https://huggingface.co/black-forest-labs/FLUX.1-dev
[7]https://huggingface.co/Wan-AI/Wan2.1-T2V-14B
[8]https://huggingface.co/Wan-AI/Wan2.2-T2V-A14B
[9]https://huggingface.co/Qwen/Qwen-Image

## C Entropy Analysis of GenAI Models

Table 4: Exponent entropy of GenAI models before and after the Hadamard transform. The evaluated models span diverse architectures (LLMs and DiTs) and modalities (language, code, image, and video), with weights drawn from both native FP8 and quantized FP8 formats. Across all models, exponent entropy remains consistently low (1.52–2.73 bits), demonstrating that exponent concentration is a universal property of trained networks. Moreover, applying the Hadamard transform does not materially alter the exponent entropy.

| Model | Model Type | FP8 Source | Exponent Entropy (Original → Hadamard) |
|---|---|---|---|
| DeepSeek-R1-0528 | LLM (MoE) | Native FP8 | 2.6037 → 2.5529 |
| Qwen3-235B-A22B-Instruct-2507-FP8 | LLM (MoE) | Native FP8 | 2.6264 → 2.6259 |
| Qwen3-Coder-30B-A3B-Instruct-FP8 | LLM (MoE) | Native FP8 | 2.5986 → 2.5979 |
| Llama-3.3-70B-Instruct-FP8-dynamic | LLM | Quantized FP8 | 2.6564 → 2.6249 |
| Qwen3-8B-FP8 | LLM | Native FP8 | 2.7295 → 2.6368 |
| FLUX.1-dev | Image DiT | Quantized FP8 | 2.0850 → 2.0861 |
| Wan2.1-T2V-14B | Video DiT | Quantized FP8 | 1.6529 → 1.6494 |
| Wan2.2-T2V-A14B | Video DiT (MoE) | Quantized FP8 | 1.5199 → 1.5178 |
| Qwen-Image | Image DiT | Quantized FP8 | 1.9503 → 1.9504 |

Our study evaluates exponent concentration across a broad set of FP8 generative models, including dense and MoE LLMs as well as image and video or image DiTs. Figure 1 shows the block wise entropy distributions, and Table 4 summarizes the entropy values for each model together with its type and FP8 source. Across all architectures, exponent entropy remains low. Large LLMs concentrate around 2.5–2.7 bits, while diffusion transformers reach values of near 1.5–2.1 bits. Post training quantized FP8 models exhibit the same pattern as native FP8 models, and the Hadamard transform leaves the exponent entropy essentially unchanged.

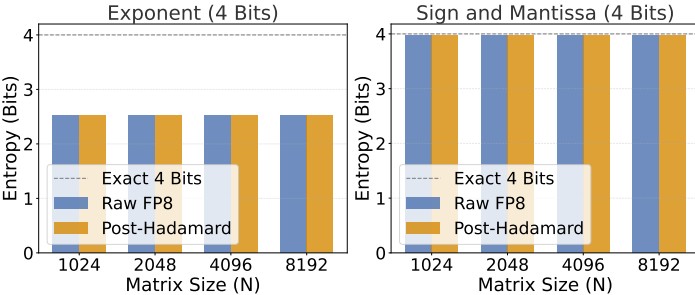

Figure 4: Exponent and sign-mantissa entropy analysis of FP8 weights sampled from $\mathcal{N}(0, 1)$.

We also visualize the exponent and sign-mantissa entropy of FP8 weights sampled from $\mathcal{N}(0, 1)$ in Figure 4. Sign and mantissa have almost exactly 4 bits of entropy regardless of whether they are Hadamard transformed or not. By comparison, the exponent entropy of FP8 weights sampled from $\mathcal{N}(0, 1)$ is around 2.6 bits, which is consistent with our observations on real models and our theoretical analysis.

# D  NUMERICAL VERIFICATION OF LOSSLESS COMPRESSION

We explicitly verify that ECF8 is strictly lossless through numerical evaluation. For image generation Figure 3, the pixel-wise difference between outputs from the FP8 model and its ECF8-compressed counterpart is exactly zero. Furthermore, we evaluated Qwen3-8B-FP8 on standard language benchmarks, finding that ECF8 and FP8 produce identical outputs. Table 5 confirms that both formats achieve the same scores across BoolQ (Clark et al., 2019), GSM8K (Cobbe et al., 2021), PIQA (Bisk et al., 2020), and Winogrande (Sakaguchi et al., 2021).

Table 5: Numerical verification of lossless compression for Qwen3-8B-FP8. ECF8 and FP8 yield identical scores across all tasks, confirming bit-perfect reconstruction.

| Benchmark | Metric | FP8 | ECF8 |
|---|---|---|---|
| GSM8K | Exact Match | $0.876 \pm 0.009$ | $0.876 \pm 0.009$ |
| BoolQ | Accuracy | $0.867 \pm 0.006$ | $0.867 \pm 0.006$ |
| PIQA | Accuracy | $0.764 \pm 0.010$ | $0.764 \pm 0.010$ |
| Winogrande | Accuracy | $0.691 \pm 0.013$ | $0.691 \pm 0.013$ |

# E  KERNEL-LEVEL BENCHMARK OF ECF8 DECOMPRESSION

In this section, we measure the absolute throughput and latency of the ECF8 decompression kernel across different amounts of FP8 weights. In H100, ECF8 decompression achieves 201–235 GB/s effective throughput and producing only 39–2177 µs of latency depending on the tensor size. The FP8 tensors used in Table 6 are generated by sampling from a normal distribution $\mathcal{N}(0, 1)$.

Table 6: Speed of the ECF8 decompression kernel on H100 GPU across different tensor sizes.

| Size (MB) | Elements (M) | Decompression Latency (µs) | Decompression Throughput (GB/s) |
|---|---|---|---|
| 8 | 8.4 | 39.6 | 201.8 |
| 32 | 33.6 | 143.3 | 223.3 |
| 64 | 67.1 | 279.1 | 229.3 |
| 128 | 134.2 | 550.4 | 232.6 |
| 256 | 268.4 | 1092.9 | 234.2 |
| 512 | 536.9 | 2177.1 | 235.2 |

We also measure per-layer latency for Qwen3-8B-FP8 comparing native FP8 and ECF8 formats. As shown in Table 7, ECF8 decoding overhead decreases as batch size increases due to higher arithmetic intensity that amortizes the decompression cost. At small batch sizes, overhead ranges from 52% (at batch size 1–64), but drops rapidly to 3% at batch size 16384. Decompression becomes negligible once GEMMs dominate the forward pass.

Table 7: Per-layer latency of Qwen3-8B-FP8 comparing native FP8 and ECF8 formats. ECF8 decoding overhead decreases as batch size increases due to higher arithmetic intensity, which amortizes the decompression cost.

| Batch Size | FP8 Latency (ms) | ECF8 Latency (ms) | Overhead (%) |
|---|---|---|---|
| 1 | 0.76 | 1.57 | 51.86 |
| 16 | 0.74 | 1.55 | 52.51 |
| 32 | 0.78 | 1.59 | 51.18 |
| 64 | 0.74 | 1.55 | 52.54 |
| 128 | 1.54 | 2.36 | 34.61 |
| 256 | 1.48 | 2.29 | 35.59 |
| 512 | 1.70 | 2.52 | 32.37 |
| 1024 | 2.03 | 2.84 | 28.70 |
| 2048 | 3.48 | 4.30 | 18.97 |
| 4096 | 6.56 | 7.37 | 11.06 |
| 8192 | 12.83 | 13.65 | 5.97 |
| 16384 | 25.17 | 25.99 | 3.14 |

To provide a more comprehensive kernel benchmark, we compare ECF8 and FP8 inference on three representative layer sizes, including a dense linear layer with 583.5M parameters from DeepSeek-R1-0528, a MoE layer with 44.0M parameters from DeepSeek-R1-0528, and a dense linear layer with 193.0M parameters from Qwen3-8B-FP8.

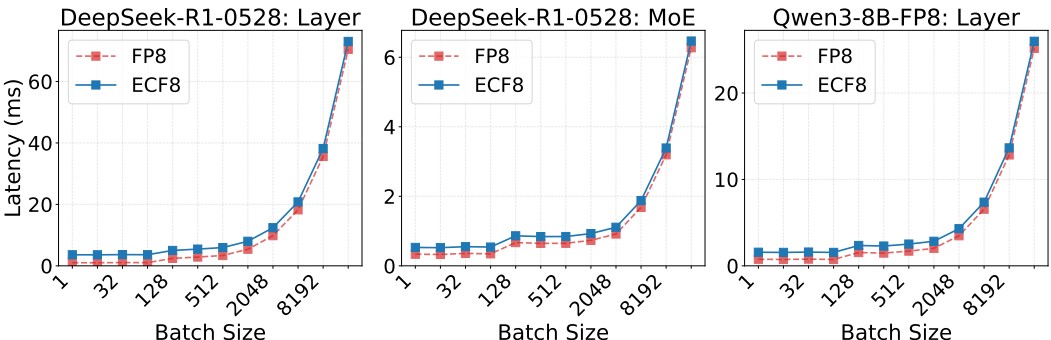

Figure 5: Layer-wise inference latency of ECF8 and FP8 versus batch size on an H100 GPU.

Figure 5 compares the layer-wise inference latency of ECF8 and FP8. For all layer sizes, the latency of ECF8 reaches increasingly closer to FP8 as batch size grows and eventually almost overlaps with FP8 when batch size is greater than 8192.

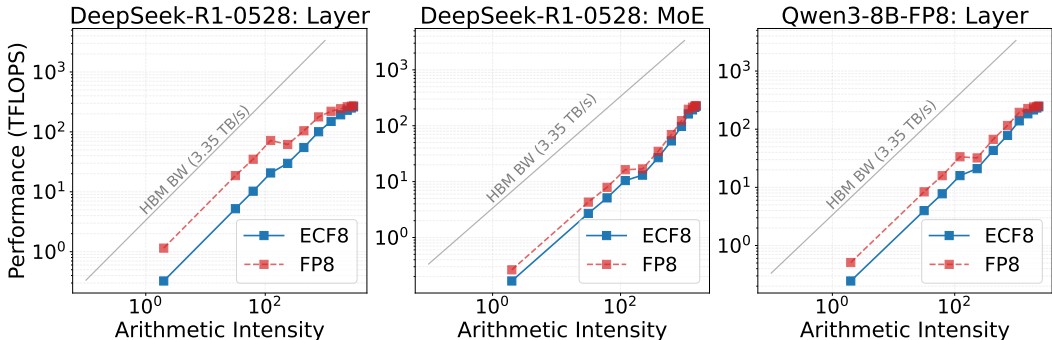

Figure 6: Layer-wise roofline analysis of ECF8 and FP8 inference on H100 GPU.

Figure 6 illustrates the roofline analysis for ECF8 versus native FP8. In low arithmetic intensity regions (memory-bound), ECF8 incurs a slight throughput penalty due to the additional decompression steps. However, as arithmetic intensity increases, driven by larger batch sizes or sequence lengths, execution becomes compute-bound. In these regimes, the decompression overhead is almost amortized by matrix multiplication, allowing ECF8 to match the peak TFLOPS of native FP8 inference.

## F    THROUGHPUT COMPARISON BETWEEN ECF8 AND FP8 IN LLMS AT THE SAME BATCH SIZE

Figure 7 presents an end-to-end throughput comparison between ECF8 and standard FP8 across multiple LLMs at matched batch sizes. ECF8 introduces a modest ∼20% throughput penalty due to the lightweight on-the-fly decompression performed before each transformer block. However, this overhead is more than compensated by ECF8's substantial memory savings: under a fixed memory budget, it can typically sustain nearly twice the batch size of FP8, as shown in Table 2. Once the workload enters the compute-bound regime, the decompression cost is fully amortized by increased arithmetic intensity, enabling ECF8 to match or even exceed FP8 throughput, consistent with the results in Table 3.

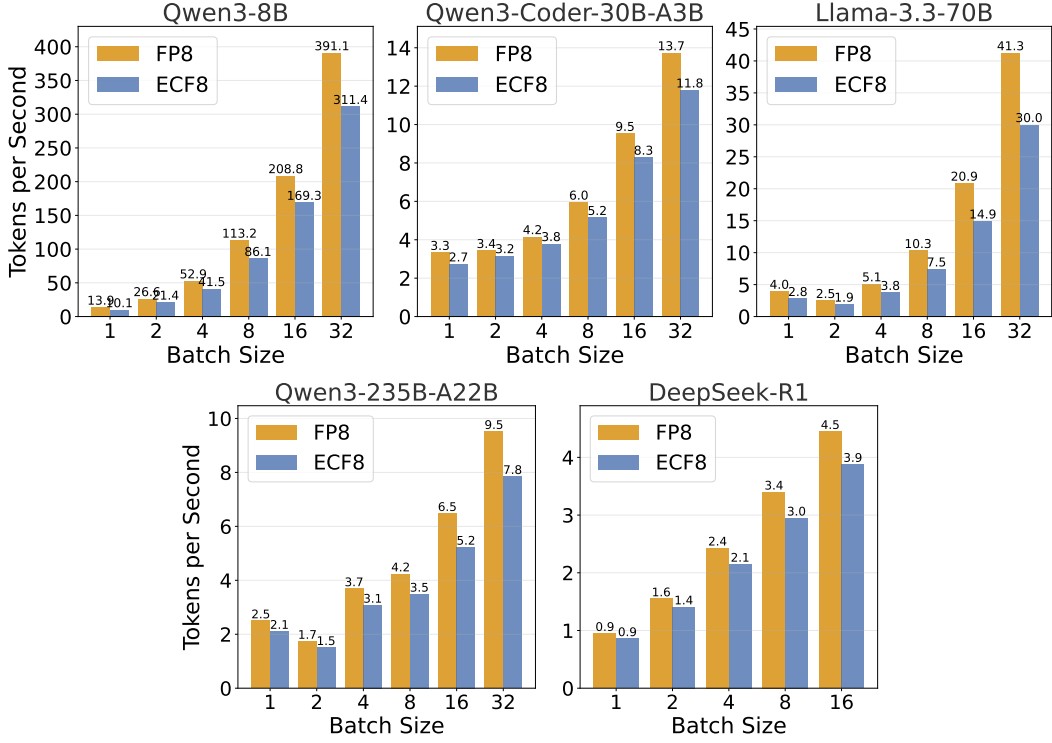

Figure 7: Throughput comparison between ECF8 and FP8 in LLMs at matched batch sizes.

## G  STEPS TO INTEGRATE ECF8 INTO VLLM OR OTHER INFERENCE FRAMEWORKS

Integrating ECF8 into production inference frameworks such as vLLM requires only small, localized modifications because ECF8 is a plug-and-play design.

- **Custom weight loader.** The framework loads compressed ECF8 weights directly into GPU memory rather than loading standard FP8 tensors. The loader reads the entropy coded exponent stream and the packed mantissa and sign bits.
- **On-demand decompression.** A small forward hook is inserted at the beginning of each transformer block. This hook invokes our GPU optimized decoder and materializes a standard FP8 tensor in a preallocated workspace buffer.

Once decompressed, the tensor is an ordinary FP8 weight array and is indistinguishable from a conventionally loaded tensor. All existing computational kernels, scheduling logic, attention implementations, and pipeline/data parallel execution paths remain unchanged.

## H  ECF8 IS COMPATIBLE WITH LORA FINE-TUNING

ECF8 is intentionally designed for static weights. Training time weights change every step, so repeatedly compressing them is impractical. Each optimizer update would require regenerating the Huffman codebook and rewriting the bitstream, which introduces overhead incompatible with training throughput. For this reason, ECF8 is positioned as an inference oriented method where weights remain fixed and can be stored once in a compact entropy coded form.

However, ECF8 is fully compatible with fine-tuning based on LoRA (Hu et al., 2022). The memory footprint of a model is dominated by the base weights, while LoRA adapters are comparatively small. In a LoRA setting the large base weights can remain in the compressed ECF8 format, while

the small LoRA adapters stay uncompressed and trainable. This preserves the standard LoRA work-flow and provides meaningful memory savings without modifying training logic.

# I   RUNTIME MEASUREMENTS FOR ECF8 ENCODING

We benchmark the speed of the ECF8 encoder and observe a sustained throughput of approximately 3.2 MB/s across various tensor sizes. Because encoding is a one-time offline preprocessing step performed prior to model release, this overhead does not impact inference latency. Table 8 shows the encoding throughput for various tensor sizes.

Table 8: ECF8 encoding throughput for various tensor sizes.

| Size (MB) | Elements (M) | Encoding Throughput (MB/s) |
|-----------|--------------|----------------------------|
| 8 | 8.4 | 3.20 |
| 32 | 33.6 | 3.17 |
| 64 | 67.1 | 3.19 |
| 128 | 134.2 | 3.18 |
| 256 | 268.4 | 3.20 |
| 512 | 536.9 | 3.08 |

We also report end-to-end encoding time for full models using 32 parallel CPU workers. Even with an unoptimized CPU encoder, the largest 671B model is compressed in under 2 hours, while smaller models take only minutes. Given that encoding is a one-time cost amortized over millions of inference runs, this overhead is negligible for deployment.

Table 9: End-to-end ECF8 encoding time for full models using 32 CPU workers.

| Model | Original Size (GB) | Encoding Time (Hours, 32 Workers) |
|-------|--------------------|-----------------------------------|
| DeepSeek-R1-0528 | 623.19 | 1.73 |
| Qwen3-235B-A22B-Instruct-2507-FP8 | 217.77 | 0.60 |
| Qwen3-Coder-30B-A3B-Instruct-FP8 | 27.85 | 0.08 |
| Llama-3.3-70B-Instruct-FP8-dynamic | 63.76 | 0.18 |
| Qwen3-8B-FP8 | 6.47 | 0.02 |
| FLUX.1-dev | 10.52 | 0.03 |
| Wan2.1-T2V-14B | 17.40 | 0.05 |
| Wan2.2-T2V-A14B | 30.49 | 0.08 |
| Qwen-Image | 26.20 | 0.07 |

## J    Software and Hardware Used in Experiments

Our experiments are conducted using PyTorch 2.7.1, CUDA 12.8, Transformers 4.56.0, and Diffusers 0.34.0. The hardware configuration varies based on model size requirements: we employ 8×H200 GPUs (141 GB memory each) for DeepSeek-R1-0528 due to its substantial memory demands, 4×H200 GPUs (141 GB memory each) for Qwen3-235B-A22B-Instruct-2507-FP8, and a single GH200 GPU (96 GB memory) for all remaining models in our evaluation suite.

## K    Image Generation Settings

We use the open-source DiffSynth implementation[10] for image generation. All DiT models are run with the default settings for image/video resolution, inference steps, and guidance scale. A fixed random seed (2025) is applied to guarantee comparability between the FP8 and ECF8 models. The prompts employed are listed in Table 10. The images generated by the ECF8 Qwen-Image model are presented in Figure 3.

Table 10: Prompts used for image generation.

| Prompt |
| --- |
| A futuristic neon-lit cityscape with a cheerful cyberpunk character in a glowing high-tech exosuit, smiling with holographic tattoos and a reflective visor. The atmosphere is bright and vibrant with neon blues and purples, blending anime and sci-fi concept art aesthetics. Holding a sign saying ECF8 IS FAST AND LOSSLESS. |
| A playful surrealist artwork where colorful balloons float through a sunny meadow, and a joyful faceless figure relaxes in midair. The palette is light and cheerful with splashes of gold and pastel tones, evoking a sense of carefree happiness. Holding a sign saying ECF8 IS FAST AND LOSSLESS. |
| An anime female character, lofi style, soft colors, gentle natural linework, key art, emotion is happy. Hand drawn with an award-winning anime aesthetic and a well-defined nose. Holding a sign saying ECF8 IS FAST AND LOSSLESS. |
| A festive parade scene with a vibrant character at the center of a confetti-filled street, smiling brightly, with balloons and streamers in the background. Holding a sign saying ECF8 IS FAST AND LOSSLESS. |
| A radiant anime-style character standing in a glowing crystal meadow, surrounded by rainbows and magical sparkles, smiling with pure happiness. Holding a sign saying ECF8 IS FAST AND LOSSLESS. |
| A cheerful fantasy scene where a character rides a friendly dragon while flying a kite, filled with vibrant colors and joyful energy. Holding a sign saying ECF8 IS FAST AND LOSSLESS. |
| A pastel illustration of a character sitting by a glowing campfire, with warm lanterns floating above, smiling peacefully. Holding a sign saying ECF8 IS FAST AND LOSSLESS. |
| A joyful carnival sunset with a character in front of a Ferris wheel, holding cotton candy, illuminated by golden evening light. Holding a sign saying ECF8 IS FAST AND LOSSLESS. |
| A futuristic festival scene where a neon-clad character dances joyfully under holographic fireworks. Holding a sign saying ECF8 IS FAST AND LOSSLESS. |
| A serene beach scene with a character building a sandcastle under a rainbow, while dolphins leap in the distance. Holding a sign saying ECF8 IS FAST AND LOSSLESS. |

---

[10]https://github.com/modelscope/DiffSynth-Studio

# L    NOTATIONS

Table 11: Symbols in Algorithm 1.

| Symbol | Description | Type / Shape |
|---|---|---|
| $LUT$ | Hierarchical lookup tables | $n_{\text{luts}} \times 256$ integers |
| $encoded$ | Encoded byte stream | $n_{\text{bytes}}$ bytes |
| $packed$ | Packed sign/mantissa nibbles | $\lceil n_{\text{elem}}/2 \rceil$ bytes |
| $outpos$ | Block output positions | $(n_{\text{blocks}}+1)$ 64-bit integers |
| $gaps$ | Packed 4-bit gap values (two per byte), across all threads | $\lceil (n_{\text{blocks}} \cdot T)/2 \rceil$ bytes |
| $L$ | 64-bit sliding bit window | 64-bit integer |
| $S$ | 16-bit tail buffer | 16-bit integer |
| $f$ | Free bits in headroom | 8-bit integer |
| $c$ | Symbols counted per thread | 32-bit integer |
| $o_{\text{start}}$ | Thread output start index | 32-bit integer |
| $o_{\text{end}}$ | Thread output end index | 32-bit integer |
| $o_{\text{base}}$ | Block output base index | 32-bit integer |
| $localbf$ | Thread register buffer | Bytes of length $B + 2$ |
| $writebf$ | Shared write buffer | $outpos[b + 1] - outpos[b]$ bytes |
| $accum$ | Shared prefix-sum array | $T + 1$ 32-bit integers |
| $n_{\text{luts}}$ | Number of lookup tables | 32-bit integer |
| $n_{\text{bytes}}$ | Length of encoded stream | 32-bit integer |
| $n_{\text{elem}}$ | Number of output elements | 32-bit integer |
| $B$ | Bytes per thread window | Constant integer |
| $T$ | Number of threads per block | Constant integer |
| $outputs$ | Decoded FP8 output bytes | $n_{\text{elem}}$ bytes |
| $b$ | Thread-block index | 32-bit integer |
| $t$ | Thread index within block | 32-bit integer |
| $t_g$ | Global thread index $(b \cdot T + t)$ | 64-bit integer |
| $g$ | Per-thread gap (extracted from $gaps$) | 4-bit integer |
| $x$ | Decoded symbol (exponent code) | 8-bit integer |
| $q$ | The packed sign/mantissa nibble | 4-bit integer |
| $b_\ell$ | Bit-length of the codeword for $x$ | 8-bit integer |

# M    PSEUDOCODE FOR ECF8 DECOMPRESSION

Table 11 summarizes the notations used in this section.

---

**Algorithm 1** Block-level decompression from ECF8 to FP8

---

**Require:** $LUT$, $encoded$, $packed$, $outpos$, $gaps$, $n_{\text{luts}}$, $n_{\text{bytes}}$, $n_{\text{elem}}$
**Ensure:** $outputs$

---

1: **Parallel for each thread** $t$ in block $b$ of size $T$:
2: Global thread id $t_g \leftarrow b \cdot T + t$.
3: Load $B + 2 = 10$ bytes from $encoded$ at offset $t_g \cdot B$ into local buffer $localbf$.
4: Form a 64-bit head $L$ from $localbf[0..7]$ so the oldest byte is the most significant; form a 16-bit tail $S$ from $localbf[8..9]$.
5: Extract 4-bit gap $g$ from packed $gaps$ using $t_g$:  $g \leftarrow \big(gaps[\lfloor t_g/2 \rfloor] \gg (4 - (t_g \bmod 2) \cdot 4)\big) \,\&\, 0x0f$. Set $L \leftarrow L \ll g$, $f \leftarrow g$, $c \leftarrow 0$.
   **Phase 1: symbol counting**
6: **while** $f < 16$ **do**
7:     $x \leftarrow LUT[L \gg 56]$
8:     **if** $x \geq 240$ **then**   $x \leftarrow LUT\big[256(256 - x) + ((L \gg 48) \,\&\, 255)\big]$
9:     **end if**
10:     $b_\ell \leftarrow LUT\big[256(n_{\text{luts}} - 1) + x\big]$; $L \leftarrow L \ll b_\ell$; $f \leftarrow f + b_\ell$; $c \leftarrow c + 1$
11: **end while**
12: Stitch tail: $L \leftarrow L \lor (S \ll (f - 16))$; $f \leftarrow f - 16$.
13: **while** $2 + \lfloor f/8 \rfloor < B$ **do**
14:     Decode one symbol as above; update $L, f, c$.
15: **end while**
   **Block-level prefix sum**
16: Each thread writes its count $c$ to shared array $accum[t]$; thread 0 writes $accum[0] \leftarrow outpos[b] + c$.
17: Perform in-place up-sweep and down-sweep on $accum[0..T-1]$ to obtain exclusive starts.
18: Thread 0 sets $accum[0] \leftarrow outpos[b]$ and $accum[T] \leftarrow outpos[b+1]$.
19: Set $o_{\text{base}} \leftarrow accum[0]$, $o_{\text{start}} \leftarrow accum[t]$, $o_{\text{end}} \leftarrow \min(o_{\text{start}} + c, n_{\text{elem}})$.
   **Phase 2: decode and assemble FP8**
20: Reinitialize $L, S, f$.
21: **while** $f < 16$ **and** $o_{\text{start}} < o_{\text{end}}$ **do**
22:     Decode next symbol $x$.
23:     $q \leftarrow packed[\lfloor o_{\text{start}}/2 \rfloor] \ll \big((o_{\text{start}} \bmod 2) \cdot 4\big)$.
24:     $byte \leftarrow (x \ll 3) \lor (q\,\&\,0x80) \lor ((q \gg 4)\,\&\,0x07)$.
25:     $writebf[o_{\text{start}} - o_{\text{base}}] \leftarrow byte$.
26:     $o_{\text{start}} \leftarrow o_{\text{start}} + 1$; update $L, f$ using $b_\ell = LUT[256(n_{\text{luts}}-1) + x]$.
27: **end while**
28: Stitch remaining: $L \leftarrow L \lor (S \ll (f - 16))$; $f \leftarrow f - 16$.
29: **while** $o_{\text{start}} < o_{\text{end}}$ **do**
30:     Decode and pack as above to fill $writebf$.
31: **end while**
   **Output write-back**
32: Synchronize all threads.
33: Each thread copies its slice of $writebf$ to $outputs$.

---

Our ECF8 decompression kernel maps each stage to the fastest GPU memory tiers. The Huffman bitstream state is maintained entirely in registers, minimizing global memory traffic. Shared memory handles block-wide prefix sums and serves as a temporary buffer for coalesced global writes. Hierarchical Huffman tables and packed mantissa data are accessed via the read-only path and cached in L2 for low-latency lookups. As a result, the decompression loop executes almost entirely out of registers and L2, with shared memory coordinating writes, enabling high sustained throughput.

