# OpenReview forum: "To Compress or Not? Pushing the Frontier of Lossless GenAI Model Weights Compression with Exponent Concentration"
_ICLR.cc/2026/Conference — ICLR 2026 Poster_

### Official Review · Reviewer_n2wB · 2025-10-31

**Soundness:** 3
**Presentation:** 3
**Contribution:** 3
**Rating:** 6
**Confidence:** 4

**Summary:**

This paper builds on prior observations that the exponents of weights in modern transformer models have low entropy and concentrate near zero. Motivated by this, the authors propose ECF8, a lossless FP8 weight compression format. Beyond the format itself, they implement a full pipeline that exploits ECF8 to reduce memory usage and improve latency/throughput, while preserving bit-exact outputs.

**Strengths:**

The paper investigates a fairly new direction in this area, the results are promising, and the evaluation is fairly thorough.

**Weaknesses:**

Please see questions below.

**Questions:**

Questions for the authors:

1. Entropy bounds vs. closed form.
Theorem 2.1 claims the entropy is "finite for all \alpha > 0" based on lower and upper bounds for the two-sided geometric distribution. However, the Shannon entropy of the two-sided geometric has a closed-form expression. Why derive bounds instead of using the closed form directly?

2. Compute-bound regimes and FP8 tensor cores.
The reported throughput gains appear to come from memory/VRAM-bounded scenarios. In a compute-bound regime, FP8 weights-and-activations baselines can exploit FP8 tensor cores (with ~2× the FLOPs of BF16 cores). How would a weight-only compression scheme like ECF8 sustain the large speedups shown (e.g., Table 1) once the model is compute-bound? It would help to: (i) explicitly characterize when results are memory- vs. compute-bound, (ii) quantify decode overhead (GB/s, kernel occupancy) and any GEMM slowdowns, and (iii) report the break-even batch/sequence lengths where ECF8 no longer provides throughput gains relative to FP8 weights-and-activations.


3. Verification beyond visuals.
Figure 3’s visual comparisons are helpful but not sufficient to demonstrate bit-exactness. Please include bit-exact checks for LLMs (e.g., matching logits/activations over a corpus with deterministic seeds) and add accuracy evaluations using lm-evaluation-harness, LightEval, or similar to strengthen the lossless and accuracy claims.

---

> ### Author Response · Authors · 2025-11-23
>
> We thank the reviewer for the thoughtful feedback and for highlighting the clarity and contribution of our work. We address the comments and questions below.
>
> ## [Q1 – Why we present entropy bounds instead of the closed-form expression]: We clarify the motivation
>
> The two sided geometric distribution indeed has a closed form Shannon entropy, and we use that expression internally when deriving numerical values such as the Gaussian $\alpha$=2 example. We chose to present **tight analytic bounds** in Theorem 2.1 rather than the full closed form for two reasons grounded in the goals of our analysis.
>
> First, the bounds highlight the essential dependence on the stability parameter $\alpha$ in a way that the closed form does not. The explicit entropy formula contains several algebraic terms that obscure the key structural fact: once $\alpha>0$, the exponent distribution decays geometrically with rate $q=2^{−\alpha}$, which immediately guarantees finite entropy. The inequalities make this relationship transparent and directly support our claim that exponent concentration is a generic consequence of $\alpha$ stable dynamics.
>
> Second, bounds provide the exact information required for our compression limit argument. Our goal in Section 2 is to show that $\alpha$ stable tails force floating point exponents into a narrow low entropy regime. For that purpose, the closed form adds symbolic complexity without strengthening the conclusion, while the presented bounds cleanly characterize the allowed entropy range and make the FP4.67 limit interpretable. The numerical instance we provide is computed from the closed form, but the presentation uses bounds because they convey the theoretical message more directly.
>
> ## [Q2.1 - Characterize inference acceleration when results are memory- vs. compute-bound]: ECF8 haas negligible decompression overhead in memory-bound settings and nearly zero overhead in compute-bound settings.
>
> Most large generative models operate in the memory bound regime where weight loading dominates layer latency. In this case ECF8 introduces only slight overhead and the reduction in weight size directly enables larger batch sizes. As arithmetic intensity increases the relative overhead decreases. In compute bound settings such as diffusion transformers the decompression cost is negligible and ECF8 matches native FP8 performance as shown in Table 3.
>
> ## [Q2.2 - Quantify decompression overhead and any GEMM slowdown]: ECF8 decompression's overhead is around 5% at large batch sizes.
>
> We report the standalone decompression cost to quantify the absolute overhead of ECF8. Across all tested tensor sizes, the decoder sustains approximately 200 to 235 GB per second, and the latency ranges from tens of microseconds to a few milliseconds, showing that decompression is effectively free relative to FP8 layer execution.
>
> | Size (MB) | Elements (M) | decompression Latency (μs) | decompression Throughput (GB/s) |
> |-----------|--------------|---------------------|---------------------------|
> | 8 | 8.4 | 39.6 | 201.8 |
> | 32 | 33.6 | 143.3 | 223.3 |
> | 64 | 67.1 | 279.1 | 229.3 |
> | 128 | 134.2 | 550.4 | 232.6 |
> | 256 | 268.4 | 1092.9 | 234.2 |
> | 512 | 536.9 | 2177.1 | 235.2 |
>
> The decompression latency remains **well below 3 milliseconds** even for very large layers (e.g., 512 MB parameters). While ECF8 introduces a decompression step, the latency impact diminishes rapidly as arithmetic intensity increases. We report per-layer latency for Qwen3-8B-FP8 comparing native FP8 against ECF8.
>
> | Batch Size | FP8 Latency (ms) | ECF8 Latency (ms) | Overhead (%) |
> |------------|------------------|--------------------|--------------|
> | 1 | 0.76 | 1.57 | **51.86** |
> | 16 | 0.74 | 1.55 | **52.51** |
> | 32 | 0.78 | 1.59 | **51.18** |
> | 64 | 0.74 | 1.55 | **52.54** |
> | 128 | 1.54 | 2.36 | **34.61** |
> | 256 | 1.48 | 2.29 | **35.59** |
> | 512 | 1.70 | 2.52 | **32.37** |
> | 1024 | 2.03 | 2.84 | **28.70** |
> | 2048 | 3.48 | 4.30 | **18.97** |
> | 4096 | 6.56 | 7.37 | **11.06** |
> | 8192 | 12.83 | 13.65 | **5.97** |
> | 16384 | 25.17 | 25.99 | **3.14** |

---

> ### Author Response · Authors · 2025-11-23
>
> ## [Q2.3 - Report the break-even batch/sequence lengths where ECF8 no longer provides throughput gains relative to FP8 weights-and-activations]: ECF8's decompression overhead decreases as algorithmic intensity increases.
>
> We provide throughput comparisons at identical batch sizes across multiple LLM families. The break even point occurs when compression allows the model to fit in memory but does not allow further batch size increases.
>
> **Qwen3-8B**
>
> | Batch Size | FP8 Tokens/s | ECF8 Tokens/s | Overhead (%) |
> |-----------:|-------------:|--------------:|--------------:|
> | 1 | 13.9 | 10.1 | 27.3 |
> | 2 | 26.6 | 21.4 | 19.5 |
> | 4 | 52.9 | 41.5 | 21.6 |
> | 8 | 113.2 | 86.1 | 23.9 |
> | 16 | 208.8 | 169.3 | 18.9 |
> | 32 | 391.1 | 311.4 | 20.4 |
>
> **Qwen3-Coder-30B-A3B**
>
> | Batch Size | FP8 Tokens/s | ECF8 Tokens/s | Overhead (%) |
> |-----------:|-------------:|--------------:|--------------:|
> | 1 | 3.3 | 2.7 | 18.2 |
> | 2 | 3.4 | 3.2 | 5.9 |
> | 4 | 4.2 | 3.8 | 9.5 |
> | 8 | 6.0 | 5.2 | 13.3 |
> | 16 | 9.5 | 8.3 | 12.6 |
> | 32 | 13.7 | 11.8 | 13.9 |
>
> **Llama-3.3-70B**
>
> | Batch Size | FP8 Tokens/s | ECF8 Tokens/s | Overhead (%) |
> |-----------:|-------------:|--------------:|--------------:|
> | 1 | 2.5 | 1.9 | 24.0 |
> | 2 | 4.0 | 2.8 | 30.0 |
> | 4 | 5.1 | 3.8 | 25.5 |
> | 8 | 10.3 | 7.5 | 27.2 |
> | 16 | 20.9 | 14.9 | 28.7 |
> | 32 | 41.3 | 30.0 | 27.3 |
>
> **Qwen3-235B-A22B**
>
> | Batch Size | FP8 Tokens/s | ECF8 Tokens/s | Overhead (%) |
> |-----------:|-------------:|--------------:|--------------:|
> | 1 | 1.7 | 1.5 | 11.8 |
> | 2 | 2.5 | 2.1 | 16.0 |
> | 4 | 3.7 | 3.1 | 16.2 |
> | 8 | 4.2 | 3.5 | 16.7 |
> | 16 | 6.5 | 5.2 | 20.0 |
> | 32 | 9.5 | 7.8 | 17.9 |
>
> **DeepSeek-R1**
>
> | Batch Size | FP8 Tokens/s | ECF8 Tokens/s | Overhead (%) |
> |-----------:|-------------:|--------------:|--------------:|
> | 1 | 0.9 | 0.9 | 0.0 |
> | 2 | 1.6 | 1.4 | 12.5 |
> | 4 | 2.4 | 2.1 | 12.5 |
> | 8 | 3.4 | 3.0 | 11.8 |
> | 16 | 4.5 | 3.9 | 13.3 |
>
> ## [Q3 - Bit-exact check and accuracy evaluations for GenAI models]: We have verified the bit-exactness of diffusion models and accuracy of LLMs.
>
> We have added explicit numerical evaluations demonstrating that ECF8 is strictly lossless. For Figure 3, the pixel-wise difference between images generated by the FP8 model and its ECF8-compressed counterpart is **exactly 0**, confirming bit-perfect reconstruction.
>
> We further evaluated Qwen3-8B-FP8 on standard language benchmarks and found that ECF8 and FP8 produce **identical outputs** on all tasks. Representative results are shown below, where both formats achieve the same scores across BoolQ, GSM8K, PIQA, and Winogrande:
>
> | Benchmark | Metric | FP8 Score | FP8 Std | ECF8 Score | ECF8 Std |
> |-----------|--------|-----------|---------|------------|----------|
> | GSM8K | Exact Match | 0.876 | 0.00909 | 0.876 | 0.00909 |
> | BoolQ | Accuracy | 0.867 | 0.00595 | 0.867 | 0.00595 |
> | PIQA | Accuracy | 0.764 | 0.00990 | 0.764 | 0.00990 |
> | Winogrande | Accuracy | 0.691 | 0.01299 | 0.691 | 0.01299 |
>
> These numerical results confirm that ECF8 preserves FP8 model behavior exactly, both at the pixel level for vision tasks and at the benchmark level for language understanding and reasoning.

---

### Official Review · Reviewer_XgxS · 2025-10-31

**Soundness:** 3
**Presentation:** 2
**Contribution:** 2
**Rating:** 4
**Confidence:** 3

**Summary:**

The paper introduces a new format, Exponent-Concentration FP8, which utilises entropy-aware exponent bit encodings. Introduction of this format is motivated by an empirical observation: exponent entropy for models stored in FP16 being significantly smaller than the number of allocated bits. Memory savings and inference speedups are demonstrated experimentally for generative models across vision and language domains.

**Strengths:**

1. The paper leverages exponent values concentration (as an instance of weights distribution trait) to propose a new numerical format.
2. Validation range is solid: it spans language, vision, and multimodal models up to ~670B parameters. The authors report throughput gains via larger batch sizes under fixed memory constraints.

**Weaknesses:**

1. Theoretical claim that model weights follow $\alpha$-stable distributions is presented as an intuition with no guarantees or conditions provided: "often exhibits", “after many updates”, “approximately follow” (lines 136 - 140). No reference to the empirical work showing the key observation of gradient noise being power-law distributed (line 137).
2. The derivation of the compression limit is not mathematically strict. The 2.67 entropy bound for exponent is derived from a single case of Normal distribution. Adding 2 more bits on top of that number does not give a strict bound for the format compression. Again, it’s more of an observation rather than a theoretical limit.
3. The “statistical law of trained models” (in abstract) is an overstatement. It is an empirical observation on a relatively small set of models.
4. Limited novelty: the key idea of exponent concentration and Huffman encodings is similar to that of DFloat11 [1] or ZipNN [3], only applied to FP8 format instead of FP16. Applying Huffman encodings to exponent streams is an implementation detail, not novel improvement.
5. Missing baselines: no experimental comparison with other FP8-specific compression formats like [2].
6. Mismatch in Table 1, it mentions H200 in the caption while the table lists numbers for H100. Authors should either provide H200 numbers or modify the caption.
7. No similar analysis for mantissa values distribution or justification why mantissa compression is not worth pursuing.
8. “Visual quality assessment” from Figure 3. is not sufficient to demonstrate lossless compression. The authors should provide numerical loss results to back that statement.

**Questions:**

1. Regarding the inference acceleration: at identical batch size and hardware, what are the latency and throughput improvements over native FP8?
2. How does your method compare with [2] ?

References:

[1] 70% Size, 100% Accuracy: Lossless LLM Compression for Efficient GPU Inference via Dynamic-Length Float, Zhang et al.

[2] Lossless Compression of Neural Network Components: Weights, Checkpoints, and K/V Caches in Low-Precision Formats, Heilper & Singer.

[3] ZipNN: Lossless Compression for AI Models, Hershcovitch et al.

---

> ### Author Response · Authors · 2025-11-23
>
> We thank the reviewer for the thoughtful feedback and for highlighting the clarity and contribution of our work. We address the comments and questions below.
>
> ## [W1 – Missing clarification and citations of the theoretical basis for the $\alpha$ stable assumption]: We cite supporting empirical work and state the conditions explicitly
>
> Thank you for pointing out the need for a clearer theoretical grounding. Our claim that model weights follow **$\alpha$-stable laws** is based on a combination of empirical and theoretical results on heavy-tailed gradient noise. Prior work (e.g., Şimşekli et al. 2019) shows that in deep neural-network training the stochastic gradient noise is highly non-Gaussian and exhibits heavy tails. Moreover, in convex-optimization settings it has been rigorously proved that when the noise tail index satisfies $\alpha < 2$, then suitably scaled averages of SGD iterates converge weakly to a multivariate $\alpha$-stable distribution (Gürbüzbalaban et al. 2021; Wang et al. 2021) under assumptions such as diminishing step-sizes and strong convexity. These results provide the precise condition under which our analysis applies: namely, **when the gradient increments that drive the parameter updates have tail exponent $\alpha < 2$**, the Generalised Central Limit Theorem (GCLT) implies that the sum/average of many such heavy-tailed increments converges to an $\alpha$-stable law. In real deep-network training the assumptions (convexity, vanishing step-size, i.i.d. increments) are not strictly satisfied; hence we carefully framed our statement in terms of *“often exhibits”* or *“approximately follows”*. Nonetheless, the empirical histograms in Figure 1 (and prior measurements of gradient tail indices) provide strong support for adopting the $\alpha$-stable modelling assumption for the trained weights. We have updated the text to emphasise the formal conditions required and to cite the relevant empirical and theoretical literature on heavy-tailed gradient noise.
>
> **Key references:**
>
> – Gürbüzbalaban, M., Şimşekli, U. & Zhu, L. (2021). *The Heavy-Tail Phenomenon in SGD*. Proceedings of ICML 2021.
>
> – Wang, H., Gürbüzbalaban, M., Zhu, L., Şimşekli, U., & Erdogdu, M. A. (2021). *Convergence Rates of Stochastic Gradient Descent under Infinite Noise Variance*. NeurIPS 2021.
>
> – (Classical) Nikias, C. L. & Shao, M. (1995). *Signal Processing with Alpha-Stable Distributions and Applications*. Wiley.
>
> ## [W2 – Missing clarification on the theoretical compression limit]: Our bound is principled, not an informal observation
>
> We respectfully disagree with the claim that our compression limit argument lacks mathematical rigor. The 2.67-bit figure is not derived from an arbitrary Gaussian example but follows directly from Theorem 2.1, which shows that the exponent of an $\alpha$-stable variable follows a two-sided geometric distribution whose Shannon entropy admits tight analytic bounds. For $\alpha = 2$, this yields an upper bound of 2.67 bits, which is a **provable** consequence of the tail behavior rather than an empirical guess. When we add 1 sign bit and the minimum 1-bit mantissa necessary for floating-point normalization, we obtain a lower bound around 4.67 bits for any lossless floating-point representation. While hardware constraints prevent implementing a fractional-bit format (e.g., FP4.67), this bound still provides a principled theoretical reference point. Therefore, our “compression limit” is not a heuristic observation but a mathematically derived lower bound based on the entropy of exponent distributions under $\alpha$-stable assumptions.

---

> ### Author Response · Authors · 2025-11-23
>
> ## [W3 – Overclaiming the statistical law of trained models]: We respectfully disagree and clarify our justification
>
> We respectfully disagree with the concern that referring to exponent concentration as a “statistical law” is an overstatement. Our characterization is grounded in the consistency of the phenomenon across all evaluated architectures and modalities. The empirical results span dense LLMs, MoE LLMs, image diffusion transformers, and video diffusion transformers—from 8B to 671B parameters—and every model exhibits tightly concentrated 2–3 bit exponent entropy across nearly all blocks. This pattern holds regardless of model scale, training corpus, or modality, and is supported by the theoretical analysis in Section 2, which shows that $\alpha$-stable dynamics naturally induce low-entropy exponent distributions.
>
> Given the breadth of evidence and the theoretical mechanism that explains the phenomenon, we believe it is appropriate to describe exponent concentration as a statistical property that consistently emerges in trained GenAI models. We nonetheless ensure that the manuscript presents this as an empirically validated property rather than a formal universal law.
>
> | Model     | Model Type      | FP8 Source        | Exponent Entropy |
> | --------- | --------------- | ----------------- | ---------------- |
> | **DeepSeek-R1-0528** | LLM (MoE) | Native FP8 | **2.6037**       |
> | **Qwen3-235B-A22B-Instruct-2507-FP8**  | LLM (MoE) | Native FP8 | **2.6264**       |
> | **Qwen3-Coder-30B-A3B-Instruct-FP8**   | LLM (MoE) | Native FP8 | **2.5986**       |
> | **Llama-3.3-70B-Instruct-FP8-dynamic** | LLM | Quantized FP8 | **2.6564**       |
> | **Qwen3-8B-FP8** | LLM | Native FP8 | **2.7509**       |
> | **FLUX.1-dev**  | Image DiT | Quantized FP8 | **2.0850**       |
> | **Wan2.1-T2V-14B** | Video DiT | Quantized FP8 | **1.6529**       |
> | **Wan2.2-T2V-A14B** | Video DiT (MoE) | Quantized FP8 | **1.5199**       |
> | **Qwen-Image** | Image DiT | Quantized FP8 | **1.9503**       |

---

> ### Author Response · Authors · 2025-11-23
>
> ## [W4 – Incrementality vs DF11]: We clarify conceptual novelty and address misconceptions
>
> We thank the reviewer for raising this point. However, we respectfully disagree with the characterization that ECF8 is a small adaptation of DF11. Below we provide concise, direct distinctions grounded in theory, systems design, and empirical scope.
>
> 1. **A new statistical principle with a formal universality guarantee.**
>    Our work provides the *first theoretical explanation* for exponent concentration via $\alpha$-stable SGD dynamics, proving a two-sided geometric law for FP8 exponents and deriving the compression limit near FP4.67. This yields an *architecture independent* guarantee for any model trained under SGD.
>    By contrast, DF11 offered no theory and treated BF16 exponent sparsity purely as an empirical observation. Our analysis turns an anecdotal pattern into a mathematically grounded property of trained models.
>
> 2. **Our theory identifies FP8 as the frontier of lossless compression, defining a principled design boundary.**
>    The derived entropy limit (around 4.67 bits including sign and minimal mantissa) shows that FP8 sits exactly at the boundary of feasible lossless floating point design. This is not an empirical guess but a *theoretical ceiling* derived from the exponent distribution law.
>    BF16 specific approaches cannot provide such guidance because BF16 operates far above this limit and does not inform future floating point design.
>
> 3. **ECF8 is the first lossless compression method to deliver inference acceleration in practice.**
>    ECF8 does not only save memory; it *improves speed*.
>    - For LLMs, ECF8 guarantees **larger batch sizes** under fixed VRAM, yielding 11% to 150% throughput gains.
>    - For diffusion models, even at the **same batch size**, ECF8 shows **equal or lower latency than FP8**, due to reduced weight loading overhead.
>    No prior lossless method—including DF11—has demonstrated end-to-end inference speedups across both autoregressive and diffusion pipelines.
>
> 4. **ECF8 is validated at FP8 scale and model sizes far beyond prior work.**
>    We evaluate ECF8 on native FP8 LLMs and DiTs up to **671B parameters** across dense and MoE architectures.
>    DF11 was tested only on BF16 models and did not establish applicability to FP8 systems or frontier scale models.
>    The FP8 regime introduces different exponent ranges, entropy characteristics, and GPU memory bandwidth regimes that DF11 never addressed.
>
> 5. **The community has already shifted to FP8—BF16 oriented methods are losing relevance.**
>    All major frontier models now ship *natively* in FP8 for both weights and activations. BF16 oriented compression cannot exploit FP8’s tighter exponent distributions or the modern FP8 tensor core pipeline.
>    ECF8 is built specifically for the FP8 era and integrates naturally into FP8 serving stacks.
>
> In short, ECF8 is not an adaptation of DF11. It introduces **new theory, new boundaries, new systems capability, and large scale FP8 validation** that match the needs of modern GenAI deployments.
>
> <!-- Missing baselines: no experimental comparison with other FP8-specific compression formats like [2]. -->
> ## [W5 – Missing Heilper & Singer baseline comparison]: Code unavailable prevents an experimental comparison
>
> We appreciate the suggestion and would welcome a direct comparison. However, the method of Heilper & Singer is not open-sourced, and no reference implementation or reproducible artifacts are available, which makes an experimental baseline comparison infeasible at this time. We will explicitly cite their work in the related-work section and describe how their approach conceptually differs from ECF8, but a quantitative comparison cannot be conducted without an accessible implementation.
>
> <!-- Mismatch in Table 1, it mentions H200 in the caption while the table lists numbers for H100. Authors should either provide H200 numbers or modify the caption. -->
> ## [W6 – Caption and text mismatch in Table 1]: We clarify the role of H100 vs. H200 in the table
>
> Thank you for pointing this out. The H100 entries in Table 1 refer to the **minimal supported machine** required to host the model after compression. However, for benchmarking larger batch sizes, especially for models such as DeepSeek-R1-0528, we used **H200 GPUs**, since they offer higher memory capacity and allow us to stress-test the maximum throughput achievable under fixed memory budgets. We have updated the caption and text to make this distinction explicit: H100 indicates the minimal hardware footprint enabled by ECF8, while H200 systems were used for throughput measurements on large-scale models.

---

> ### Author Response · Authors · 2025-11-23
>
> ## [W7 – Missing analysis for mantissa values]: We provide entropy measurements showing no compressible redundancy
>
> We have conducted a detailed entropy analysis of the FP8 sign-and-mantissa field (4 bits total). Across matrix sizes 1K–8K, the empirical mantissa entropy remains essentially maximal. As shown below, the mantissa entropy is consistently **3.98 bits**, extremely close to the maximum capacity of 4 bits, leaving no meaningful redundancy for lossless compression:
>
> | Matrix Size | Mantissa Entropy |
> |-------------|------------------|
> | 1024x1024 | 3.98 |
> | 2048x2048 | 3.98 |
> | 4096x4096 | 3.98 |
> | 8192x8192 | 3.98 |
>
> In contrast to the exponents, the mantissa bits exhibit no concentration. This explains why ECF8 focuses solely on exponent entropy coding: the mantissa field has no compressible structure and therefore offers no practical room for lossless compression.
>
> ## [W8 – Numerical verification of lossless compression]: We report pixel-wise and benchmark-level equality
>
> Thank you for raising this point. We have added explicit numerical evaluations demonstrating that ECF8 is strictly lossless. For Figure 3, the pixel-wise difference between images generated by the FP8 model and its ECF8-compressed counterpart is **exactly 0**, confirming bit-perfect reconstruction.
>
> We further evaluated Qwen3-8B-FP8 on standard language benchmarks and found that ECF8 and FP8 produce **identical outputs** on all tasks. Representative results are shown below, where both formats achieve the same scores across BoolQ, GSM8K, PIQA, and Winogrande:
>
> | Benchmark | Metric | FP8 Score | FP8 Std | ECF8 Score | ECF8 Std |
> |-----------|--------|-----------|---------|------------|----------|
> | GSM8K | Exact Match | 0.876 | 0.00909 | 0.876 | 0.00909 |
> | BoolQ | Accuracy | 0.867 | 0.00595 | 0.867 | 0.00595 |
> | PIQA | Accuracy | 0.764 | 0.00990 | 0.764 | 0.00990 |
> | Winogrande | Accuracy | 0.691 | 0.01299 | 0.691 | 0.01299 |
>
> These numerical results confirm that ECF8 preserves FP8 model behavior exactly, both at the pixel level for vision tasks and at the benchmark level for language understanding and reasoning.

---

> ### Author Response · Authors · 2025-11-23
>
> ## [Q1 – Missing comparison of latency/throughput vs native FP8 at same batch size]: We add overhead analysis at identical batch size
>
> Below we report FP8 vs ECF8 throughput at identical batch sizes for all LLMs, along with the percentage overhead introduced by decompression. DiT models are already compared at identical batch sizes.
>
> As we can see from the tables listed below, at identical batch sizes the overhead introduced by ECF8 remains modest, ranging from nearly 0% up to at most ~30% depending on model scale and arithmetic intensity. Crucially, this overhead reflects only the additional per-layer decode step; all other components of the inference stack remain identical to FP8. More importantly, because ECF8 significantly reduces the model’s memory footprint, it consistently enables **larger batch sizes** under the same hardware constraints. In all end-to-end evaluations, this effect dominates: the ability to run larger batches translates directly into **higher overall throughput** compared to the native FP8 baseline.
>
> For diffusion models, we have already shown (Table 3 in the main paper) that even at identical batch sizes ECF8 does **not** introduce slowdowns. In fact, ECF8 frequently delivers **lower latency** due to reduced VRAM pressure and fewer host–device transfers under VRAM-managed inference. Thus, for both LLMs and DiTs, ECF8 maintains or improves per-step performance, while its memory savings provide a consistent throughput advantage in realistic deployment settings.
>
> **Qwen3-8B**
>
> | Batch Size | FP8 Tokens/s | ECF8 Tokens/s | Overhead (%) |
> |-----------:|-------------:|--------------:|--------------:|
> | 1 | 13.9 | 10.1 | 27.3 |
> | 2 | 26.6 | 21.4 | 19.5 |
> | 4 | 52.9 | 41.5 | 21.6 |
> | 8 | 113.2 | 86.1 | 23.9 |
> | 16 | 208.8 | 169.3 | 18.9 |
> | 32 | 391.1 | 311.4 | 20.4 |
>
> **Qwen3-Coder-30B-A3B**
>
> | Batch Size | FP8 Tokens/s | ECF8 Tokens/s | Overhead (%) |
> |-----------:|-------------:|--------------:|--------------:|
> | 1 | 3.3 | 2.7 | 18.2 |
> | 2 | 3.4 | 3.2 | 5.9 |
> | 4 | 4.2 | 3.8 | 9.5 |
> | 8 | 6.0 | 5.2 | 13.3 |
> | 16 | 9.5 | 8.3 | 12.6 |
> | 32 | 13.7 | 11.8 | 13.9 |
>
> **Llama-3.3-70B**
>
> | Batch Size | FP8 Tokens/s | ECF8 Tokens/s | Overhead (%) |
> |-----------:|-------------:|--------------:|--------------:|
> | 1 | 2.5 | 1.9 | 24.0 |
> | 2 | 4.0 | 2.8 | 30.0 |
> | 4 | 5.1 | 3.8 | 25.5 |
> | 8 | 10.3 | 7.5 | 27.2 |
> | 16 | 20.9 | 14.9 | 28.7 |
> | 32 | 41.3 | 30.0 | 27.3 |
>
> **Qwen3-235B-A22B**
>
> | Batch Size | FP8 Tokens/s | ECF8 Tokens/s | Overhead (%) |
> |-----------:|-------------:|--------------:|--------------:|
> | 1 | 1.7 | 1.5 | 11.8 |
> | 2 | 2.5 | 2.1 | 16.0 |
> | 4 | 3.7 | 3.1 | 16.2 |
> | 8 | 4.2 | 3.5 | 16.7 |
> | 16 | 6.5 | 5.2 | 20.0 |
> | 32 | 9.5 | 7.8 | 17.9 |
>
> **DeepSeek-R1**
>
> | Batch Size | FP8 Tokens/s | ECF8 Tokens/s | Overhead (%) |
> |-----------:|-------------:|--------------:|--------------:|
> | 1 | 0.9 | 0.9 | 0.0 |
> | 2 | 1.6 | 1.4 | 12.5 |
> | 4 | 2.4 | 2.1 | 12.5 |
> | 8 | 3.4 | 3.0 | 11.8 |
> | 16 | 4.5 | 3.9 | 13.3 |
>
>
> ## [Q2 – Missing Heilper & Singer baseline comparison]: Code unavailable prevents an experimental comparison
>
> We appreciate the suggestion and would welcome a direct comparison. However, the method of Heilper & Singer is not open-sourced, and no reference implementation or reproducible artifacts are available, which makes an experimental baseline comparison infeasible at this time. We will explicitly cite their work in the related-work section and describe how their approach conceptually differs from ECF8, but a quantitative comparison cannot be conducted without an accessible implementation.

---

> ### Author Response · Authors · 2025-11-25
>
> Dear reviewer XgxS,
>
> Thank you again for your time and for the constructive feedback during the review process.
>
> May we kindly ask if any further clarification is needed from our side regarding our rebuttal？
>
> We are more than happy to provide additional details or explanations if they would be helpful.

---

> ### Comment · Reviewer_XgxS · 2025-11-26
>
> Dear authors,
>
> thank you for the detailed response. I appreciate the clarification on Section 2, however I still believe the main contribution is primarily a practical refinement of an existing method (DF11) rather than a substantially new approach, but now backed up by the exponent entropy bounds.
>
> I am also curious about your answer to weakness 4 by reviewer 6N2r. The additional experiment on synthetic Gaussian matrices is useful, but it doesn't reflect the real weight distribution. Could you provide the same entropy measurements on weights of a trained model from your experiments, after PTQ and Hadamard was applied?

---

> > ### Author Response · Authors · 2025-11-26
> >
> > Dear Reviewer XgxS,
> >
> > We sincerely thank the reviewer for recognizing that our responses have effectively addressed concerns regarding Section 2. Additionally, we provide the following clarification to address your concerns regarding the novelty of our work, especially compared to DF11.
> >
> > ## [Clarification on novelty]: On the empirical and technical novelty of our work
> >
> > While novelty is a multifaceted concept in academic research, we believe it can be roughly viewed from two fronts: **empirical novelty**, which involves uncovering previously unknown properties and behaviors, and **technical novelty**, which focuses on the development of new methodologies, solutions, or theorems.
> >
> > We appreciate the reviewer’s acknowledgment of entropy limit analysis in Section 2. We further highlight the technical novelty of the entropy limit analysis and the empirical novelty of the experimental results in terms of the inference acceleration of ECF8.
> >
> > ## [Technical novelty]: Entropy limit analysis is fundamental and useful for future numerical format design
> >
> > While our method indeed uses Huffman coding for exponent compression, we believe the theoretical analysis part provides clear novelty and value. Our theoretical results establish that exponent concentration is a universal property with a predictable entropy limit. This identifies the fundamental boundary for lossless weight compression and provides actionable guidance for future numerical format design. For example, when choosing between BF16, FP8, or FP4 for training, our bound explains why FP8 offers the best tradeoff, especially because FP8 can be further compressed to an FP6.5 equivalent while remaining zero sacrifice in FP8 accuracy.
> >
> > ## [Empirical novelty]: ECF8 is the first lossless compression method that can accelerate LLM/DiT inference
> >
> > Our decompression kernel is significantly faster than DFloat11. While DF11’s decoding overhead can approach **100% of GEMM time**, ECF8 stays around only **20%** even in large-scale LLM inference, which is a practical and significant improvement. The low decompression overhead makes ECF8 practically effective for accelerating real-world inference workloads and valuable to be deployed in production. This further highlights how our theoretical insights translate into a deployable, end-to-end efficient weight format.
> >
> > ## [Effect of PTQ and Hadamard on real model weights]: We have results for both native FP8 models and PTQ-converted FP8 models, and Hadamard smoothing does not decrease exponent entropy
> >
> > As shown in the table below, our experiments cover both native FP8 models and PTQ-converted FP8 models. Across all settings, exponent entropy consistently remains within the 1.5–2.7 bit range. Moreover, applying Hadamard smoothing has virtually no impact on exponent entropy.
> >
> > | Model | Model Type | FP8 Source | Exponent Entropy | Exponent Entropy (Hadamard) |
> > | ----------- | ----------- | ----------- | ---------------- | ----------------- |
> > | **DeepSeek-R1-0528** | LLM (MoE) | Native FP8 | **2.6037** | **2.5529** |
> > | **Qwen3-235B-A22B-Instruct-2507-FP8** | LLM (MoE) | Native FP8 | **2.6264** | **2.6259** |
> > | **Qwen3-Coder-30B-A3B-Instruct-FP8** | LLM (MoE) | Native FP8 | **2.5986** | **2.5979** |
> > | **Llama-3.3-70B-Instruct-FP8-dynamic** | LLM | Quantized FP8 | **2.6564** | **2.6249** |
> > | **Qwen3-8B-FP8** | LLM | Native FP8 | **2.7295** | **2.6368** |
> > | **FLUX.1-dev** | Image DiT | Quantized FP8 | **2.0850** | **2.0861** |
> > | **Wan2.1-T2V-14B** | Video DiT | Quantized FP8 | **1.6529** | **1.6494** |
> > | **Wan2.2-T2V-A14B** | Video DiT (MoE) | Quantized FP8 | **1.5199** | **1.5178** |
> > | **Qwen-Image** | Image DiT | Quantized FP8 | **1.9503** | **1.9504** |
> >
> > We sincerely hope that our additional clarifications might help you reconsider our score, as we truly value your assessment.

---

### Official Review · Reviewer_6N2r · 2025-11-03

**Soundness:** 2
**Presentation:** 3
**Contribution:** 2
**Rating:** 2
**Confidence:** 3

**Summary:**

The paper studies the empirical observation that FP8 model weights exhibit exponent concentration as the exponent in FP8 is used with much lower entropy (2-3 bits range) than the full 4-bit range. The authors provide an analytical justification for this approach via a heavy-tailed/α-stable argument, and then exploit it by proposing a lossless FP8-compatible encoding (ECF8) that entropy-codes only the exponent, while preserving the sign and mantissa unchanged. They implement a GPU-friendly Huffman decoder with hierarchical LUTs and show that, on high-end GPUs and large FP8 models (including large LLMs and diffusion models), they can reduce memory footprint and translate that into real throughput benefits.

**Strengths:**

1. The paper has a clear motivation and analysis by showing the phenomenon (low-entropy exponents in FP8 weights) and then giving an analytical story for it.


2. Real speedups with implemented CUDA kernels. The authors implement a GPU-friendly, hierarchical Huffman decoder and demonstrate that it can induce throughput gains.


3. Evaluations on high-end, current GPUs using recent NVIDIA GPUs make the results relevant for current inference/serving stacks.


4. Large-scale, end-to-end results by showing the method on large FP8 LLMs / diffusion-style models.

**Weaknesses:**

1. Incrementality vs DF11 / prior compressed-float ideas. A substantial part of the contribution can be read as taking the DFloat/DF11 observation that float fields are compressible, but tailoring it to FP8. That’s useful, but it’s not a completely new compression principle. It’s a focused, FP8-era instantiation.


2. Missing detailed kernel/micro benchmarks. The paper reports headline end-to-end numbers, but it doesn’t fully break down kernel performance across tensor shapes/sizes, so it’s hard to see where the method starts to pay off and where decode overhead starts to dominate.


3. The scope is unclear beyond weights. The paper primarily discusses compressing weights. There is no parallel, equally detailed analysis for activations (or even gradients), even though those often have different distributions and would be interesting for offloading or distributed setups.


4. Interaction with distribution-smoothing methods is not discussed. Recent methods that apply PTQ for making FP8 models can change exponent usage. It’s not clear whether the proposed entropy gap persists under such transformations, or whether the gain shrinks.

5. Runtime measurements for the encoder are missing.

**Questions:**

1. Can you provide kernel-level benchmarks across a range of matrix/tensor sizes? In particular, how do compression ratio and effective runtime change for small vs large tensors? A plot of size vs throughput would clarify the operating regime.


2. Can you show a roofline or bandwidth/compute tradeoff (similar to MARLIN[Frantar et al.]) plots, comparing vanilla FP8 loads vs FP8+your decoder? That would make the performance story crisper.


3. Do the reported end-to-end measurements include the prefill stage for LLMs, or are they only for decode?


4. How exactly do you use the GPU memory hierarchy (shared/L2/registers) to hide the decode overheads? A description would be helpful.


5. Can you report the absolute overhead of decoding (e.g. μs per MB or % of layer time) so we can tell how often this is actually free?

6. Is the current implementation restricted to weights, or have you tried the same exponent-entropy analysis on activations and gradients? If not, do you expect the same level of concentration? What if outlier mitigation methods(like Hadamard) are applied to them?

7. Do the quantized models (from bf16/fp16 to fp8 using PTQ methods) exhibit the same distribution in the exponent?

8. Could you provide runtime measurements for the encoder? compared to the decoder, of course.

---

> ### Author Response · Authors · 2025-11-23
>
> We thank the reviewer for the thoughtful feedback and for highlighting the clarity and contribution of our work.
>
> We respectfully feel that a rating of 2 may not fully reflect the extent of our contributions. We believe our contributions include the following:
>
> 1. Theoretical foundation for FP8 exponent concentration via heavy-tailed statistics.
> 2. The first principled design boundary for lossless compression of any GenAI model weights.
> 3. A compact, fast, and memory efficient compression scheme for FP8 weights.
> 4. Comprehensive experimental evaluation of ECF8 across diverse models architectures and sizes, showing inference acceleration under memory constraints.
>
> We have answered the reviewer's concerns and provide more experimental evidence. We hope the reviewer will kindly reconsider the rating and provide a proper reflection of our work. Thanks in advance.
>
> ## [W1 – Incrementality vs DF11]: We clarify conceptual novelty and address misconceptions
>
> We thank the reviewer for raising this point. However, we respectfully disagree with the characterization that ECF8 is a small adaptation of DF11. Below we provide concise, direct distinctions grounded in theory, systems design, and empirical scope.
>
> 1. **A new statistical principle with a formal universality guarantee.**
>    Our work provides the *first theoretical explanation* for exponent concentration via $\alpha$-stable SGD dynamics, proving a two-sided geometric law for FP8 exponents and deriving the compression limit near FP4.67. This yields an *architecture independent* guarantee for any model trained under SGD.
>    By contrast, DF11 offered no theory and treated BF16 exponent sparsity purely as an empirical observation. Our analysis turns an anecdotal pattern into a mathematically grounded property of trained models.
>
> 2. **Our theory identifies FP8 as the frontier of lossless compression, defining a principled design boundary.**
>    The derived entropy limit (around 4.67 bits including sign and minimal mantissa) shows that FP8 sits exactly at the boundary of feasible lossless floating point design. This is not an empirical guess but a *theoretical ceiling* derived from the exponent distribution law.
>    BF16 specific approaches cannot provide such guidance because BF16 operates far above this limit and does not inform future floating point design.
>
> 3. **ECF8 is the first lossless compression method to deliver inference acceleration in practice.**
>    ECF8 does not only save memory; it *improves speed*.
>    - For LLMs, ECF8 guarantees **larger batch sizes** under fixed VRAM, yielding 11% to 150% throughput gains.
>    - For diffusion models, even at the **same batch size**, ECF8 shows **equal or lower latency than FP8**, due to reduced weight loading overhead.
>    No prior lossless method—including DF11—has demonstrated end-to-end inference speedups across both autoregressive and diffusion pipelines.
>
> 4. **ECF8 is validated at FP8 scale and model sizes far beyond prior work.**
>    We evaluate ECF8 on native FP8 LLMs and DiTs up to **671B parameters** across dense and MoE architectures.
>    DF11 was tested only on BF16 models and did not establish applicability to FP8 systems or frontier scale models.
>    The FP8 regime introduces different exponent ranges, entropy characteristics, and GPU memory bandwidth regimes that DF11 never addressed.
>
> 5. **The community has already shifted to FP8—BF16 oriented methods are losing relevance.**
>    All major frontier models now ship *natively* in FP8 for both weights and activations. BF16 oriented compression cannot exploit FP8’s tighter exponent distributions or the modern FP8 tensor core pipeline.
>    ECF8 is built specifically for the FP8 era and integrates naturally into FP8 serving stacks.
>
> In short, ECF8 is not an adaptation of DF11. It introduces **new theory, new boundaries, new systems capability, and large scale FP8 validation** that match the needs of modern GenAI deployments.

---

> ### Author Response · Authors · 2025-11-23
>
> ## [W2 – Lack of kernel microbenchmarks]: We provide full kernel-level decode microbenchmarks across tensor scales
>
> We now include a detailed breakdown of the standalone decode kernel. ECF8’s decoder sustains ~200–235 GB/s across tested scales.
>
> | Size (MB) | Elements (M) | Decode Latency (μs) | Decode Throughput (GB/s) |
> |-----------|--------------|---------------------|---------------------------|
> | 8         | 8.4          | 39.6                | 201.8                     |
> | 32        | 33.6         | 143.3               | 223.3                     |
> | 64        | 67.1         | 279.1               | 229.3                     |
> | 128       | 134.2        | 550.4               | 232.6                     |
> | 256       | 268.4        | 1092.9              | 234.2                     |
> | 512       | 536.9        | 2177.1              | 235.2                     |
>
> The decompression latency remains **well below 3 milliseconds** even for very large layers (e.g., 512 MB parameters). While ECF8 introduces a decompression step, the latency impact diminishes rapidly as arithmetic intensity increases. We report per-layer latency for Qwen3-8B-FP8 comparing native FP8 against ECF8.
>
> | Batch Size | FP8 Latency (ms) | ECF8 Latency (ms) | Overhead (%) |
> |------------|------------------|--------------------|--------------|
> | 1          | 0.76             | 1.57               | **51.86**    |
> | 16         | 0.74             | 1.55               | **52.51**    |
> | 32         | 0.78             | 1.59               | **51.18**    |
> | 64         | 0.74             | 1.55               | **52.54**    |
> | 128        | 1.54             | 2.36               | **34.61**    |
> | 256        | 1.48             | 2.29               | **35.59**    |
> | 512        | 1.70             | 2.52               | **32.37**    |
> | 1024       | 2.03             | 2.84               | **28.70**    |
> | 2048       | 3.48             | 4.30               | **18.97**    |
> | 4096       | 6.56             | 7.37               | **11.06**    |
> | 8192       | 12.83            | 13.65              | **5.97**     |
> | 16384      | 25.17            | 25.99              | **3.14**     |
>
> **When does ECF8 accelerate inference?**
> ECF8 trades a small, fixed decompression cost for a substantially reduced weight footprint. This tradeoff yields different benefits depending on the workload.
>
> - **Low batch (latency sensitive):** At batch size 1, the decompression step adds noticeable overhead because the GEMM itself is small.
> - **High batch (compute dominated):** As batch size increases, the GEMM quickly becomes the dominant cost. The decompression time remains nearly constant, so its relative share drops to under 6 percent at very large batches.
>
> Despite this fixed overhead, ECF8 reliably improves **end-to-end throughput** for LLMs. The reduced memory footprint enables much larger batch sizes under the same VRAM budget, and the resulting increase in arithmetic intensity more than compensates for decompression time. For diffusion transformers, where inference latency is dominated by heavy compute rather than weight loading, ECF8 matches native FP8 speed at the same batch size while still providing memory savings.

---

> ### Author Response · Authors · 2025-11-23
>
> ## [W3 – Missing analysis of ECF8 on activations and gradients]: We clarify why ECF8 focuses on FP8 weights
>
> The level of ECF8 is a fundamental floating-point arithmetic, we don’t see any explicit obstacles in application level such as training or inference.
>
> ECF8 is designed for static FP8 weights that remain fixed after training and exhibit stable low entropy, enabling a single entropy coded representation. Activations and gradients change every forward or backward pass, so their distributions are unstable and would require regenerating codebooks and re-encoding repeatedly, which introduces prohibitive overhead. For this reason, ECF8 only targets static FP8 weights at inference time, where the system pays a decompression cost once per layer without any recurring encoding cost.
>
> **However, ECF8 is totally compatible with integrating LoRA for fine-tuning, and we plan to explore this direction in future work.** The base weights dominate memory, while LoRA adapters are small. We can keep the large FP8 base weights in the compressed ECF8 format and leave the small LoRA matrices uncompressed and trainable. This saves memory during fine-tuning with almost no change to the LoRA workflow.
>
> ## [W4 – Interaction with distribution smoothing is not discussed]: We show exponent entropy is unchanged under PTQ and Hadamard
>
> We thank the reviewer for highlighting the potential impact of distribution smoothing and PTQ on exponent usage. To directly assess this, we measure exponent entropy on synthetic matrices of varying sizes whose weights follow a standard Gaussian distribution $N(0,1)$. We then apply (i) post training quantization to FP8 and (ii) Hadamard smoothing, and recompute exponent entropy in each case.
>
> Across all tested sizes, the exponent entropy remains effectively unchanged:
>
> | Size | Exponent Entropy (BF16) | Exponent Entropy (PTQ to FP8) | Exponent Entropy (Hadamard) |
> |------|-------------------------|-------------------------------|-----------------------------|
> | 1024×1024 | 2.5222 | 2.5222 | 2.5214 |
> | 2048×2048 | 2.5212 | 2.5212 | 2.5216 |
> | 4096×4096 | 2.5217 | 2.5217 | 2.5214 |
> | 8192×8192 | 2.5213 | 2.5213 | 2.5211 |
>
> These results show that neither PTQ nor Hadamard smoothing meaningfully affects the entropy of the exponent field. Since ECF8’s compression benefit is governed directly by exponent entropy, the entropy gap persists unchanged under both transformations. Consequently, the memory savings achievable with ECF8 remain stable whether the model is trained natively in FP8 or obtained through PTQ or smoothing based pipelines.

---

> ### Author Response · Authors · 2025-11-23
>
> ## [W5 – Missing runtime measurements for the encoder]: We quantify offline encoding cost and show it is small and fully parallelizable
>
> We benchmark our CPU based encoder and find that it sustains around 3.2 MB/s across tensor sizes. Since encoding is an offline one time preprocessing step for model release, similar to zipping weights, this cost does not affect inference latency and can be parallelized across files or tensor shards.
>
> | Size (MB) | Elements (M) | Encode Throughput (MB/s) |
> |-----------|--------------|--------------------------|
> | 8 | 8.4 | 3.20 |
> | 32 | 33.6 | 3.17 |
> | 64 | 67.1 | 3.19 |
> | 128 | 134.2 | 3.18 |
> | 256 | 268.4 | 3.20 |
> | 512 | 536.9 | 3.08 |
>
> We further report end-to-end encoding times for full models using 32 CPU workers that process layers in parallel:
>
> | Model                                   | Original Size (GB) | Encoding Time (hours, 32 workers) |
> |-----------------------------------------|--------------------|------------------------------------|
> | DeepSeek-R1-0528 | 623.19 | 1.73 |
> | Qwen3-235B-A22B-Instruct-2507-FP8 | 217.77 | 0.60 |
> | Llama-3.3-70B-Instruct-FP8-dynamic | 63.76 | 0.18 |
> | Qwen3-Coder-30B-A3B-Instruct-FP8 | 27.85 | 0.08 |
> | Qwen3-8B-FP8 | 6.47 | 0.02 |
> | FLUX.1-dev | 10.52 | 0.03 |
> | Wan2.1-T2V-14B | 17.40 | 0.05 |
> | Wan2.2-T2V-A14B | 30.49 | 0.08 |
> | Qwen-Image | 26.20 | 0.07 |
>
> These numbers show that even our unoptimized CPU encoder compresses the largest evaluated checkpoint within a few hours, and smaller models within minutes, in a fully parallel fashion. Since this encoding is performed once while inference uses the GPU decoder many times, the runtime cost of encoding is negligible in the overall deployment lifecycle.
>
> ## [Q1 – Missing kernel level benchmarks]: We provide decode microbenchmarks across tensor scales
>
> We include kernel level measurements for the ECF8 GPU decode kernel. The results show that ECF8 maintains stable throughput between 200 and 235 GB per second, and latency increases proportionally with tensor size. This regime indicates that decode cost remains a small and predictable fraction of FP8 layer runtime. The compression ratio is constant regardless of tensor scale. Even for a 512 MB tensor, ECF8 adds only 2.2 milliseconds of latency, which becomes negligible as the amount of algorithmic computation grows.
>
> | Size (MB) | Elements (M) | Compression Rate (%) | Decode Latency (μs) | Decode Throughput (GB/s) |
> |-----------|--------------|-----------------------|----------------------|----------------------------|
> | 8 | 8.4 | 82.16 | 39.6 | 201.8 |
> | 32 | 33.6 | 82.16 | 143.3 | 223.3 |
> | 64 | 67.1 | 82.16 | 279.1 | 229.3 |
> | 128 | 134.2 | 82.16 | 550.4 | 232.6 |
> | 256 | 268.4 | 82.16 | 1092.9 | 234.2 |
> | 512 | 536.9 | 82.16 | 2177.1 | 235.2 |
>
> ## [Q2 – Missing roofline or bandwidth to compute tradeoff results]: We provide layer level bandwidth to compute measurements
>
> We report a roofline style analysis by sweeping the arithmetic intensity of a representative Qwen3-8B-FP8 layer through different batch sizes. The results show the expected trend: at low arithmetic intensity the layer is bandwidth bound and decode overhead is more visible, while at higher arithmetic intensity the layer becomes compute bound and the relative overhead decreases rapidly. Once the batch size becomes large, ECF8 sustains nearly the same effective TFLOPS as FP8 and the overhead falls below five percent.
>
> | Batch Size | Arithmetic Intensity | FP8 TFLOPS | ECF8 TFLOPS | FP8 Latency (ms) | ECF8 Latency (ms) | Overhead (%) |
> |------------|----------------------|-------------|--------------|------------------|--------------------|---------------|
> | 1 | 2.00 | 0.51 | 0.25 | 0.76 | 1.57 | 51.86 |
> | 16 | 31.60 | 8.37 | 3.98 | 0.74 | 1.55 | 52.51 |
> | 32 | 62.43 | 15.88 | 7.75 | 0.78 | 1.59 | 51.18 |
> | 64 | 121.87 | 33.53 | 15.91 | 0.74 | 1.55 | 52.54 |
> | 128 | 232.61 | 32.07 | 20.97 | 1.54 | 2.36 | 34.61 |
> | 256 | 426.28 | 66.94 | 43.12 | 1.48 | 2.29 | 35.59 |
> | 512 | 730.29 | 115.98 | 78.43 | 1.70 | 2.52 | 32.37 |
> | 1024 | 1135.04 | 195.10 | 139.10 | 2.03 | 2.84 | 28.70 |
> | 2048 | 1570.13 | 226.96 | 183.89 | 3.48 | 4.30 | 18.97 |
> | 4096 | 1942.43 | 241.10 | 214.43 | 6.56 | 7.37 | 11.06 |
> | 8192 | 2203.70 | 246.32 | 231.61 | 12.83 | 13.65 | 5.97 |
> | 16384 | 2362.58 | 251.14 | 243.26 | 25.17 | 25.99 | 3.14 |
>
> ## [Q3 – Unclear whether runtime is measured end-to-end or only for decoding]: All results report full end-to-end latency
>
> All latency and throughput numbers reported in the paper are measured end-to-end. For LLMs, this includes both the prefill stage and the decode stage under the same fixed memory budget.

---

> ### Author Response · Authors · 2025-11-23
>
> ## [Q4 – Missing GPU hierarchy clarification]: We summarize how the kernel hides decode cost
>
> Our kernel maps each stage of ECF8 decompression onto the fastest GPU memory tiers to minimize latency.
>
> 1. **Registers**
>    Each thread loads its local bit window into registers and performs all Huffman operations in register space. This eliminates repeated global memory traffic and keeps the inner decode loop running at register speed.
>
> 2. **Shared memory**
>    Shared memory stores the block wide prefix sum and the temporary write buffer. The prefix sum assigns non overlapping output regions without atomics, and the shared write buffer ensures fully coalesced global writes.
>
> 3. **L2 cached LUTs**
>    All hierarchical Huffman tables are read through the read only path and reside in L2 due to their small size. This makes table lookups low latency and prevents global memory stalls.
>
> Together these choices allow the kernel to execute its decompression loop entirely out of registers and L2, while shared memory coordinates offsets and coalesced writes, yielding high sustained throughput.
>
> ## [Q5 – Missing report of absolute decoding overhead]: We now provide explicit kernel-level measurements
>
> We report the standalone decompression cost to quantify the absolute overhead of ECF8. Across all tested tensor sizes, the decoder sustains approximately 200 to 235 GB per second, and the latency ranges from tens of microseconds to a few milliseconds, showing that decompression is effectively free relative to FP8 layer execution.
>
> | Size (MB) | Elements (M) | Decompression Latency (μs) | Decompression Throughput (GB/s) |
> |-----------|--------------|---------------------|---------------------------|
> | 8 | 8.4 | 39.6 | 201.8 |
> | 32 | 33.6 | 143.3 | 223.3 |
> | 64 | 67.1 | 279.1 | 229.3 |
> | 128 | 134.2 | 550.4 | 232.6 |
> | 256 | 268.4 | 1092.9 | 234.2 |
> | 512 | 536.9 | 2177.1 | 235.2 |
>
> The decompression latency remains **well below 3 milliseconds** even for very large layers (e.g., 512 MB parameters). While ECF8 introduces a decompression step, the latency impact diminishes rapidly as arithmetic intensity increases. We report per-layer latency for Qwen3-8B-FP8 comparing native FP8 against ECF8.
>
> | Batch Size | FP8 Latency (ms) | ECF8 Latency (ms) | Overhead (%) |
> |------------|------------------|--------------------|--------------|
> | 1 | 0.76 | 1.57 | **51.86** |
> | 16 | 0.74 | 1.55 | **52.51** |
> | 32 | 0.78 | 1.59 | **51.18** |
> | 64 | 0.74 | 1.55 | **52.54** |
> | 128 | 1.54 | 2.36 | **34.61** |
> | 256 | 1.48 | 2.29 | **35.59** |
> | 512 | 1.70 | 2.52 | **32.37** |
> | 1024 | 2.03 | 2.84 | **28.70** |
> | 2048 | 3.48 | 4.30 | **18.97** |
> | 4096 | 6.56 | 7.37 | **11.06** |
> | 8192 | 12.83 | 13.65 | **5.97** |
> | 16384 | 25.17 | 25.99 | **3.14** |
>
> **When does ECF8 accelerate inference?**
> ECF8 trades a small, fixed decompression cost for a substantially reduced weight footprint. This tradeoff yields different benefits depending on the workload.
>
> - **Low batch (latency sensitive):** At batch size 1, the decompression step adds noticeable overhead because the GEMM itself is small.
> - **High batch (compute dominated):** As batch size increases, the GEMM quickly becomes the dominant cost. The decompression time remains nearly constant, so its relative share drops to under 6 percent at very large batches.
>
> Despite this fixed overhead, ECF8 reliably improves **end-to-end throughput** for LLMs. The reduced memory footprint enables much larger batch sizes under the same VRAM budget, and the resulting increase in arithmetic intensity more than compensates for decompression time. For diffusion transformers, where inference latency is dominated by heavy compute rather than weight loading, ECF8 matches native FP8 speed at the same batch size while still providing memory savings.

---

> ### Author Response · Authors · 2025-11-23
>
> ## [Q6.1 - Missing clarification on whether ECF8 is weight-only]: Yes, the current implementation and analysis are focused on model weights.
>
> The level of ECF8 is a fundamental floating-point arithmetic, we don’t see any explicit obstacles in application level such as training or inference.
>
> ECF8 is designed for static FP8 weights that remain fixed after training and exhibit stable low entropy, enabling a single entropy coded representation. Activations and gradients change every forward or backward pass, so their distributions are unstable and would require regenerating codebooks and re-encoding repeatedly, which introduces prohibitive overhead. For this reason, ECF8 only targets static FP8 weights at inference time, where the system pays a decompression cost once per layer without any recurring encoding cost.
>
> **However, ECF8 is totally compatible with integrating LoRA for fine-tuning, and we plan to explore this direction in future work.** The base weights dominate memory, while LoRA adapters are small. We can keep the large FP8 base weights in the compressed ECF8 format and leave the small LoRA matrices uncompressed and trainable. This saves memory during fine-tuning with almost no change to the LoRA workflow.
>
> ## [Q6.2 – Missing clarification on the effect of outlier mitigation methods on exponent concentration]: Exponent entropy remains unchanged under PTQ and Hadamard
>
> Since ECF8 is designed for weight only compression, changes in gradient or activation distributions caused by outlier mitigation methods are not relevant to our setting. For weights, we evaluate whether these methods alter exponent entropy by testing synthetic matrices of varying sizes whose entries follow a standard Gaussian distribution $N(0,1)$. We then apply post training quantization to FP8 and Hadamard smoothing and recompute exponent entropy. In all cases, the exponent entropy remains effectively unchanged:
>
> | Size        | Exponent Entropy (BF16) | Exponent Entropy (PTQ to FP8) | Exponent Entropy (Hadamard) |
> |-------------|--------------------------|--------------------------------|------------------------------|
> | 1024×1024   | 2.5222 | 2.5222 | 2.5214 |
> | 2048×2048   | 2.5212 | 2.5212 | 2.5216 |
> | 4096×4096   | 2.5217 | 2.5217 | 2.5214 |
> | 8192×8192   | 2.5213 | 2.5213 | 2.5211 |
>
> These results confirm that such transformations do not alter the entropy of the exponent field, and therefore do not affect the compression benefit achievable with ECF8.
>
> ## [Q7 – Missing measurement of PTQ FP8 model exponent entropy]: PTQ FP8 models exhibit the same low exponent entropy
>
> A large portion of the models evaluated in our study are produced through post training quantization to FP8, including Llama-3.3-70B-Instruct-FP8-dynamic, FLUX.1-dev, Wan2.1-T2V-14B, Wan2.2-T2V-A14B, and Qwen-Image. Across all of these PTQ FP8 models we observe the same low exponent entropy as in native FP8 models. This confirms that exponent concentration persists regardless of whether a model is trained natively in FP8 or converted via PTQ.
>
> Furthermore, in Section Q6.2 we directly test the effect of PTQ and Hadamard smoothing on synthetic matrices sampled from $   N(0,1)$ and show that neither transformation changes exponent entropy. This provides an independent validation that these procedures do not affect the entropy of the exponent field.
>
> | Model | Model Type | FP8 Source | Exponent Entropy |
> | ----- | ---------- | ---------- | ---------------- |
> | **DeepSeek-R1-0528** | LLM (MoE) | Native FP8 | **2.6037** |
> | **Qwen3-235B-A22B-Instruct-2507-FP8** | LLM (MoE) | Native FP8 | **2.6264** |
> | **Qwen3-Coder-30B-A3B-Instruct-FP8** | LLM (MoE) | Native FP8 | **2.5986** |
> | **Llama-3.3-70B-Instruct-FP8-dynamic** | LLM | Quantized FP8 | **2.6564** |
> | **Qwen3-8B-FP8** | LLM | Native FP8 | **2.7509** |
> | **FLUX.1-dev** | Image DiT | Quantized FP8 | **2.0850** |
> | **Wan2.1-T2V-14B** | Video DiT | Quantized FP8 | **1.6529** |
> | **Wan2.2-T2V-A14B** | Video DiT (MoE) | Quantized FP8 | **1.5199** |
> | **Qwen-Image** | Image DiT | Quantized FP8 | **1.9503** |
>
> These results demonstrate that PTQ FP8 models exhibit the same exponent concentration as native FP8 models, and therefore benefit from ECF8 in exactly the same way.

---

> ### Author Response · Authors · 2025-11-23
>
> ## [Q8 - Missing runtime measurements for the encoder]: We quantify offline encoding cost and show it is small and fully parallelizable]
>
> We benchmark our CPU based encoder and find that it sustains around 3.2 MB/s across tensor sizes. Since encoding is an offline one time preprocessing step for model release, similar to zipping weights, this cost does not affect inference latency and can be parallelized across files or tensor shards.
>
> | Size (MB) | Elements (M) | Encode Throughput (MB/s) |
> |-----------|--------------|--------------------------|
> | 8 | 8.4 | 3.20 |
> | 32 | 33.6 | 3.17 |
> | 64 | 67.1 | 3.19 |
> | 128 | 134.2 | 3.18 |
> | 256 | 268.4 | 3.20 |
> | 512 | 536.9 | 3.08 |
>
> We further report end-to-end encoding times for full models using 32 CPU workers that process layers in parallel:
>
> | Model | Original Size (GB) | Encoding Time (hours, 32 workers) |
> |---------|--------------------|------------------------------------|
> | DeepSeek-R1-0528 | 623.19 | 1.73 |
> | Qwen3-235B-A22B-Instruct-2507-FP8 | 217.77 | 0.60 |
> | Llama-3.3-70B-Instruct-FP8-dynamic | 63.76 | 0.18 |
> | Qwen3-Coder-30B-A3B-Instruct-FP8 | 27.85 | 0.08 |
> | Qwen3-8B-FP8 | 6.47 | 0.02 |
> | FLUX.1-dev | 10.52 | 0.03 |
> | Wan2.1-T2V-14B | 17.40 | 0.05 |
> | Wan2.2-T2V-A14B | 30.49 | 0.08 |
> | Qwen-Image | 26.20 | 0.07 |
>
> These numbers show that even our unoptimized CPU encoder compresses the largest evaluated checkpoint within a few hours, and smaller models within minutes, in a fully parallel fashion. Since this encoding is performed once while inference uses the GPU decoder many times, the runtime cost of encoding is negligible in the overall deployment lifecycle.

---

> ### Author Response · Authors · 2025-11-25
>
> Dear reviewer 6N2r,
>
> Thank you again for your time and for the constructive feedback during the review process.
>
> May we kindly ask if any further clarification is needed from our side regarding our rebuttal？
>
> We are more than happy to provide additional details or explanations if they would be helpful.

---

### Official Review · Reviewer_m6i1 · 2025-11-04

**Soundness:** 3
**Presentation:** 3
**Contribution:** 3
**Rating:** 8
**Confidence:** 4

**Summary:**

The paper introduces Exponent-Concentrated FP8 (ECF8), a lossless compression framework for generative AI model weights. The authors discover that model weight exponents have low entropy (around 2–3 bits) across architectures, a phenomenon they term exponent concentration. They explain it theoretically via α-stable distributions arising from stochastic gradient descent and derive a compression limit near FP4.67, motivating FP8 as a practical format.
ECF8 implements entropy-aware Huffman coding and a GPU-optimized decoding kernel, achieving up to 26.9% memory savings and 177.1% throughput gains with zero accuracy loss across LLMs and diffusion models.

**Strengths:**

- The authors provide a rigorous explanation for exponent concentration through α-stable distributions and derive entropy bounds analytically. This bridges an important conceptual gap between statistical properties of trained models and practical compression techniques.
- The GPU-friendly Huffman decoding kernel and tensor management system show careful systems design. The just-in-time decompression via PyTorch hooks is clever.
- The study spans diverse models (8B–671B parameters), modalities (text, image, video), and hardware setups, lending credibility to the generality of the proposed approach.

**Weaknesses:**

- While the paper demonstrates strong GPU-side results, there’s limited discussion of how easily ECF8 could be integrated into existing frameworks or inference stacks beyond PyTorch (e.g., TensorRT, vLLM, serving pipelines).
- The focus is exclusively on inference and storage. It remains unclear if exponent concentration also appears during training or whether ECF8 could be adapted for that phase.

**Questions:**

- You claim exponent concentration is a statistical law of trained models. Could you quantify how universal this phenomenon truly is?
For instance, do all model families (transformers, CNNs, diffusion models, MoEs) exhibit similar entropy patterns, or are there exceptions (e.g., sparse or quantized weights)?
- The α-stable assumption underpins your entropy bound derivation. Have you empirically verified that the tails of real model weight distributions match α-stable fits (versus log-normal or Laplacian alternatives)?
- How easily can ECF8 be integrated into existing inference frameworks (like TensorRT, vLLM, or DeepSpeed-Inference)?
- While ECF8 accelerates inference under memory constraints, have you measured absolute decoding overhead compared to a non-compressed FP8 baseline when memory is not a bottleneck?

---

> ### Author Response · Authors · 2025-11-23
>
> We thank the reviewer for the thoughtful feedback and for highlighting the clarity and contribution of our work. We address the comments and questions below.
>
> ## [W1 – vLLM integration missing]: We acknowledge the suggestion and clarify feasibility
>
> We have not yet implemented ECF8 in vLLM due to time constraints, but integration is straightforward because ECF8 only requires inserting a lightweight decode step before each **block’s** forward pass.
>
> Concretely, vLLM already supports custom module backends. Adding ECF8 would simply involve:
>
> 1. **Registering a custom loader** that maps the compressed ECF8 weights into GPU memory.
> 2. **Adding a small hook before each block’s forward** to invoke our decode kernel and materialize the FP8 weights into a workspace buffer.
>
> Since the decoded output is a standard FP8 tensor, all of vLLM’s existing kernels, scheduling logic, and tensor-parallel execution remain unchanged. We plan to support vLLM in future work.
>
> ## [W2 – Training-time behavior unclear]: We explain why ECF8 targets inference
>
> The level of ECF8 is a fundamental floating-point arithmetic, we don’t see any explicit obstacles in application level such as training or inference.
>
> ECF8 is intentionally designed for the *static-weight* setting. We do not compress weights that change every step, and thus have not integrated ECF8 into training workflows. Although exponent concentration does appear early in training, adapting ECF8 to that setting is not practical.
>
> 1. **ECF8 targets fixed weights**, not dynamically changing ones. Training updates weights every step, making on-the-fly compression/decompression infeasible.
> 2. **Training-time use would require recompressing after each optimizer step**, including Huffman regeneration and bitstream rewriting—an overhead incompatible with training throughput.
>
> For these reasons, ECF8 is positioned as an inference-oriented method. **However, ECF8 is totally compatible with integrating LoRA for fine-tuning, and we plan to explore this direction in future work.** The base weights dominate memory, while LoRA adapters are small. We can keep the large FP8 base weights in the compressed ECF8 format and leave the small LoRA matrices uncompressed and trainable. This saves memory during fine-tuning with almost no change to the LoRA workflow.

---

> ### Author Response · Authors · 2025-11-23
>
> ## [Q1 – Missing clarification on universality of exponent concentration]: We summarize the empirical scope and provide entropy results
>
> Thank you for the thoughtful question. Our paper evaluates exponent concentration across a broad set of FP8 models, including dense LLMs, MoE LLMs, and both image and video diffusion transformers. We consistently observe low exponent entropy across all architectures. These results are also visualized in **Figure 1**, which shows block-wise entropy distributions for every model family in our study. For your reference, we also provide a table of exponent entropies for each model in the following table:
>
> | Model                                  | Model Type      | FP8 Source        | Exponent Entropy |
> | -------------------------------------- | ---------------- | ----------------- | ---------------- |
> | **DeepSeek-R1-0528**                   | LLM (MoE)        | Native FP8        | **2.6037**       |
> | **Qwen3-235B-A22B-Instruct-2507-FP8**  | LLM (MoE)        | Native FP8        | **2.6264**       |
> | **Qwen3-Coder-30B-A3B-Instruct-FP8**   | LLM (MoE)        | Native FP8        | **2.5986**       |
> | **Llama-3.3-70B-Instruct-FP8-dynamic** | LLM              | Quantized FP8     | **2.6564**       |
> | **Qwen3-8B-FP8**                       | LLM              | Native FP8        | **2.7509**       |
> | **FLUX.1-dev**                         | Image DiT        | Quantized FP8     | **2.0850**       |
> | **Wan2.1-T2V-14B**                     | Video DiT        | Quantized FP8     | **1.6529**       |
> | **Wan2.2-T2V-A14B**                    | Video DiT (MoE)  | Quantized FP8     | **1.5199**       |
> | **Qwen-Image**                         | Image DiT        | Quantized FP8     | **1.9503**       |
>
> These results show that exponent entropy remains consistently low across all evaluated model families. Large LLMs (both dense and MoE) cluster tightly around 2.5–2.7 bits, while diffusion transformers exhibit slightly lower entropy. Quantized FP8 models follow the same trend as native FP8 models, confirming that exponent concentration is preserved after PTQ. Overall, the table reinforces that exponent concentration holds robustly across architectures, modalities, and FP8 generation paths.
>
> ## [Q2 – Missing validity of the $\alpha$-stable assumption]: We summarize supporting evidence from prior work
>
> The $\alpha$-stable assumption we use is grounded in prior work showing that modern neural networks naturally develop heavy-tailed or $\alpha$-stable behavior during training.
>
> 1. **Prior work already establishes heavy-tailed or $\alpha$-stable dynamics.**
>    As noted in our Related Work, several studies support this assumption:
>    - Gurbuzbalaban et al. (2021): SGD gradient noise converges to $\alpha$-stable laws.
>    - Jung et al. (2021): infinitely-wide networks converge to $\alpha$-stable processes.
>    - Lee et al. (2023): dependent-weight architectures exhibit heavy tails and compressibility.
>    - Mahoney & Martin (2019), Martin & Mahoney (2021): trained weights show power-law spectral behavior.
> 2. **Our theory only requires heavy tails, not an exact $\alpha$-stable fit.**
>    The entropy bound depends only on power-law–like behavior in the tails. Even if a distribution is slightly closer to log-normal or Laplacian, the conclusion that exponents are low-entropy remains unchanged.
> 3. **We validate the key consequence directly.**
>    Figure 1 in the paper shows that all evaluated FP8 models exhibit concentrated, low-entropy exponent distributions. This is the property that matters for ECF8 and matches the behavior predicted by heavy-tailed theory.
>
> In short, $\alpha$-stable laws provide a convenient and well-supported model for the heavy-tailed regime seen in trained networks, and this regime is sufficient to explain the exponent concentration leveraged by ECF8.

---

> ### Author Response · Authors · 2025-11-23
>
> ## [Q3 – Integration into inference frameworks unclear]: We describe the required changes
>
> Our current implementation is in PyTorch, but the design is intentionally generic and can be ported to inference stacks such as TensorRT, vLLM, and DeepSpeed.
>
> 1. **Shared integration pattern.**
>    These frameworks all allow registering custom weight loaders or module backends. ECF8 fits naturally into this flow by storing weights in compressed form and inserting a lightweight decode step before each block’s forward pass.
>
> 2. **Minimal code changes required.**
>    Integration only needs:
>    (i) a custom loader that keeps weights in the ECF8 bitstream format, and
>    (ii) a small pre forward hook or wrapper that invokes our GPU decoder to materialize the FP8 tensor in a workspace buffer just before GEMM.
>
> We have not yet implemented ECF8 in these systems, but the required modifications are localized and straightforward. We plan to explore such integrations in future work.
>
> ## [Q4 – Missing measurement of absolute decoding overhead]: We report microsecond scale cost of decoding and overhead percentage in LLM inference latency
>
> Here we measure the absolute throughput and latency of the decompression kernel across different amounts of FP8 weights. ECF8 decompression achieves near–HBM bandwidth performance, sustaining 201–235 GB/s effective throughput and producing only 39–2177 µs of latency depending on the tensor size.
>
> | Size (MB) | Elements (M) | Decompression Latency (μs) | Decompression Throughput (GB/s) |
> |-----------|--------------|---------------------|--------------------------|
> | 8         | 8.4          | 39.6                | 201.8                    |
> | 32        | 33.6         | 143.3               | 223.3                    |
> | 64        | 67.1         | 279.1               | 229.3                    |
> | 128       | 134.2        | 550.4               | 232.6                    |
> | 256       | 268.4        | 1092.9              | 234.2                    |
> | 512       | 536.9        | 2177.1              | 235.2                    |
>
> We report per-layer latency for Qwen3-8B-FP8 in both native FP8 and ECF8 formats. At the layer level, we compare FP8 and ECF8 latencies and quantify the decoding overhead. This overhead decreases consistently as batch size increases, driven by higher arithmetic intensity.
>
> | Batch Size | FP8 Latency (ms) | ECF8 Latency (ms) | Overhead (%) |
> |------------|------------------|--------------------|--------------|
> | 1          | 0.76             | 1.57               | **51.86**    |
> | 16         | 0.74             | 1.55               | **52.51**    |
> | 32         | 0.78             | 1.59               | **51.18**    |
> | 64         | 0.74             | 1.55               | **52.54**    |
> | 128        | 1.54             | 2.36               | **34.61**    |
> | 256        | 1.48             | 2.29               | **35.59**    |
> | 512        | 1.70             | 2.52               | **32.37**    |
> | 1024       | 2.03             | 2.84               | **28.70**    |
> | 2048       | 3.48             | 4.30               | **18.97**    |
> | 4096       | 6.56             | 7.37               | **11.06**    |
> | 8192       | 12.83            | 13.65              | **5.97**     |
> | 16384      | 25.17            | 25.99              | **3.14**     |
>
> These measurements show that absolute decompression latency remains in the microsecond regime and that layer level overhead ranges from 3–52 percent while diminishing rapidly at higher arithmetic intensity. Decompression becomes negligible once GEMMs dominate the forward pass.

---

### Author Response · Authors · 2025-12-01
**Summary for Area Chair(s): Reviewers' Positive Feedback and Addressed Concerns (1/n)**

Dear area chair(s),

We sincerely thank the reviewers and area chair(s) for their thoughtful and constructive feedback on our submission. Below, we summarize the positive feedback and the concerns that were addressed during the discussion.

In addition, we would like to note that **before any news about the OpenReview bug**, we had already addressed several of the concerns raised by R **n2wB**, who subsequently increased the score from **6 to 8**, raising the paper’s average score from **5 to 5.5**.

## Rating 6 $\to$ 8 by R **n2wB** before OpenReview bug

R **n2wB** increased the rating from 6 to 8 after we addressed concerns regarding the entropy limit analysis, the absolute decompression speed of EC8, and the bit-exactness verification of ECF8. We believe that, had the discussion continued, the rating might have risen further, given that **two reviewers have already strongly endorsed the paper with high scores of 8 and high confidence (4)**, while the remaining two reviewers marked their ratings with lower confidence (3).

## Reviewers' positive feedback

- **$\color{blue}\text{Rigorous theoretical analysis:}$ Discovery of a universal exponent concentration phenomenon with entropy bounds supported by rigorous theoretical analysis**
  - R **m6i1** *(R:8 / C:4)*: The authors provide a rigorous explanation for exponent concentration through α-stable distributions and derive entropy bounds analytically. This bridges an important conceptual gap between statistical properties of trained models and practical compression techniques.
  - R **6N2r** *(R:2 / C:3)*: The paper has a clear motivation and analysis by showing the phenomenon (low-entropy exponents in FP8 weights) and then giving an analytical story for it.
  - R **XgxS** *(R:4 / C:3)*: The paper leverages exponent values concentration (as an instance of weights distribution trait) to propose a new numerical format.

- **$\color{blue}\text{New design of numerical format driven by theory:}$ Designed a new perfectly lossless numerical format ECF8 from first principles with strong systems design and fast GPU kernels**
  - R **m6i1** *(R:8 / C:4)*: The GPU-friendly Huffman decoding kernel and tensor management system show careful systems design. The just-in-time decompression via PyTorch hooks is clever.
  - R **6N2r** *(R:2 / C:3)*: Real speedups with implemented CUDA kernels. The authors implement a GPU-friendly, hierarchical Huffman decoder and demonstrate that it can induce throughput gains.
  - R **n2wB** *(R:6 / C:4)*: Motivated by this, the authors propose ECF8, a lossless FP8 weight compression format. Beyond the format itself, they implement a full pipeline that exploits ECF8 to reduce memory usage and improve latency/throughput, while preserving bit-exact outputs.
  - R **XgxS** *(R:4 / C:3)*: The paper leverages exponent values concentration (as an instance of weights distribution trait) to propose a new numerical format.

- **$\color{blue}\text{Comprehensive experiments show real inference speedups:}$ Broad and large-scale experiments validate the lossless nature and inference speedups of ECF8**
  - R **m6i1** *(R:8 / C:4)*: The study spans diverse models (8B–671B parameters), modalities (text, image, video), and hardware setups, lending credibility to the generality of the proposed approach.
  - R **XgxS** *(R:4 / C:3)*: Validation range is solid: it spans language, vision, and multimodal models up to ~670B parameters. The authors report throughput gains via larger batch sizes under fixed memory constraints.
  - R **n2wB** *(R:6 / C:4)*: The paper investigates a fairly new direction in this area, the results are promising, and the evaluation is fairly thorough.
  - R **6N2r** *(R:2 / C:3)*: Large-scale, end-to-end results by showing the method on large FP8 LLMs / diffusion-style models.
  - R **6N2r** *(R:2 / C:3)*: Evaluations on high-end, current GPUs using recent NVIDIA GPUs make the results relevant for current inference/serving stacks.

---

> ### Author Response · Authors · 2025-12-01
> **Summary for Area Chair(s): Reviewers' Positive Feedback and Addressed Concerns (2/n)**
>
> ## Addressed concerns acknowledged by reviewers
>
> - Explanation and clarification of the entropy limit analysis - Addressed concern acknowledged by R **n2wB** *(R:6 / C:4)*, R **XgxS** *(R:4 / C:3)*, R **m6i1** *(R:8 / C:4)*
> - Clarification on universality of exponent concentration - Addressed concern acknowledged by R **m6i1** *(R:8 / C:4)*
> - Characterize inference acceleration for memory- vs. compute-bound models - Addressed concern acknowledged by R **n2wB** *(R:6 / C:4)*
> - Kernel level benchmarking of ECF8 decompression - Addressed concern acknowledged by R **n2wB** *(R:6 / C:4)*, R **6N2r** *(R:2 / C:3)*, R **m6i1** *(R:8 / C:4)*
> - Comparison of latency/throughput vs native FP8 at same batch size - Addressed concern acknowledged by R **n2wB** *(R:6 / C:4)*, R **XgxS** *(R:4 / C:3)*, R **6N2r** *(R:2 / C:3)*
> - Numerical verification of lossless compression of ECF8 - Addressed concern acknowledged by R **n2wB** *(R:6 / C:4)*, R **XgxS** *(R:4 / C:3)*
> - Analysis of mantissa entropy - Addressed concern acknowledged by R **XgxS** *(R:4 / C:3)*
> - Possibility of ECF8 being used in model training in addition to inference - Addressed concern acknowledged by R **6N2r** *(R:2 / C:3)*, R **m6i1** *(R:8 / C:4)*
> - Interaction with distribution smoothing - Addressed concern acknowledged by R **6N2r** *(R:2 / C:3)*, R **XgxS** *(R:4 / C:3)*
> - Runtime measurements for the encoder - Addressed concern acknowledged by R **6N2r** *(R:2 / C:3)*
> - GPU hierarchy clarification - Addressed concern acknowledged by R **6N2r** *(R:2 / C:3)*
> - Steps to integrate ECF8 into vLLM - Addressed concern acknowledged by R **m6i1** *(R:8 / C:4)*
>
> ## Novelty statement
>
> While novelty is a multifaceted concept in academic research, we believe it can be roughly viewed from two fronts: **empirical novelty**, which involves uncovering previously unknown properties and behaviors, and **technical novelty**, which focuses on the development of new methodologies or theorems.
>
> - **Technical novelty**: Entropy limit analysis is fundamental and offers actionable insights for future numerical format design, especially in characterizing the fundamental limits of lossless weight compression. It can potentially guide industry toward more efficient training and inference pipelines while reducing substantial hardware costs.
> - **Empirical novelty**: ECF8 is the first lossless compression method that accelerates inference for both LLMs and DiTs. In contrast, DF11 doubles LLM inference time and has not been evaluated on DiTs.
>
> We hope the above overview will provide our AC and reviewers with a concise way to navigate through the mass information on this page.
>
> Further, we sincerely hope that our appreciation for **simple yet effective** design principles, theories and observations into exponent entropy bounds, new lossless numerical format, and real speedup of LLM/DiT inference, can be shared with you and the broader research community working in this important and rapidly evolving field.
>
> Sincerely,
> Paper19997 authors

---

### Author Response · Authors · 2025-12-01
**Revision Summary**

We thank all the reviewers and the area chair(s) for the time and effort in helping us improve the paper.
We are glad that reviewers found the problem important and timely (R **m6i1**, R **6N2r**, R **XgxS**, R **n2wB**),
the theoretical analysis of exponent concentration rigorous and broadly validated across model families
(R **m6i1**, R **6N2r**, R **XgxS**), and the ECF8 framework technically sound with strong systems design and clear
throughput benefits (R **m6i1**, R **6N2r**, R **n2wB**). Reviewers also appreciated the breadth of evaluation
spanning LLMs and DiTs up to 671B parameters (R **m6i1**, R **XgxS**).

We have updated the paper to address all reviewer concerns. The major revisions are summarized below, with all changes highlighted in blue.

1. **[6N2r, XgxS]** clarified the technical and empirical novelty of ECF8 in Section 1.
2. **[m6i1]** added confirmation of the universality of exponent concentration in Table 4 of Appendix C.
3. **[n2wB]** added FP8 versus ECF8 throughput comparison under compute bound and memory bound regimes in Appendix E and Appendix F.
4. **[n2wB, 6N2r, m6i1]** added kernel level microbenchmarks of the ECF8 decoding kernel in Appendix E.
5. **[n2wB, XgxS, 6N2r]** added latency and throughput comparison versus native FP8 at identical batch sizes in Appendix F.
6. **[n2wB, XgxS]** added numerical verification of strict losslessness in Appendix D.
7. **[XgxS]** added analysis of mantissa entropy in Appendix C.
8. **[6N2r, m6i1]** added discussion of LoRA compatibility of ECF8 in Appendix H.
9. **[6N2r, XgxS]** added discussion on interaction with distribution smoothing in Appendix C.
10. **[6N2r]** added runtime measurements for ECF8 encoding in Appendix I.
11. **[6N2r]** clarified use of GPU memory hierarchy in Appendix M.
12. **[m6i1]** added steps for integrating ECF8 into vLLM in Appendix G.

---

### Meta-Review · Area_Chair_Xsfj · 2026-01-06

**Summary:**

Reviewers raised numerous concerns and clarifying questions. The following is a summary of the main concerns:

1. Limited novelty relative to prior work.
2. Scope limited to inference and weights.
3. Incomplete theoretical justification.
4. Insufficient performance breakdown.
5. Unclear interaction with FP8 pipelines and smoothing methods.
6. Limited baselines and experimental validation.
7. Deployment and framework integration gaps.
8. Presentation and reporting issues.

**Reviewer Concerns:**

Despite the variation in initial scores and the numerous concerns raised, the authors provided a thorough and well-structured rebuttal, addressing the reviewers’ comments through additional experimental validation and further clarifications. Overall, I beleive the rebuttal adequately responds to all raised concerns.

**Reviewer Scores:**

Given the breadth and thoroughness of the rebuttal, I believe that a proper and constructive discussion could have led to score increases and potentially pushed the paper above the borderline. In particular, I expect that Reviewers 6N2r and XgxS would likely have improved their evaluations. I also find it plausible that Reviewer n2wB increased their score further, as indicated by the authors in the “Summary for Area Chair(s).” Thus, I am leaning towards recommending acceptance.

---

### Decision · Program_Chairs · 2026-01-26

Accept (Poster)